# MuHBoost: Multi-Label Boosting for Practical Longitudinal Human Behavior Modeling

**Nguyen Thach**[†][*]**, Patrick Habecker**[†]**, Anika Eisenbraun**[†]**, Alex Mason**[†]**,**

**Kimberly Tyler**[†]**, Bilal Khan**[§] **& Hau Chan**[†]

## Abstract

Longitudinal human behavior modeling has received increasing attention over the years due to its widespread applications to patient monitoring, dietary and lifestyle recommendations, and just-in-time intervention for at-risk individuals (e.g., problematic drug users and struggling students), to name a few. Using in-the-moment health data collected via ubiquitous devices (e.g., smartphones and smartwatches), this multidisciplinary field focuses on developing predictive models for certain health or well-being outcomes (e.g., depression and stress) in the short future given the time series of individual behaviors (e.g., resting heart rate, sleep quality, and current feelings). Yet, most existing models on these data, which we refer to as *ubiquitous health data*, do not achieve adequate accuracy. The latest works that yielded promising results have yet to consider realistic aspects of ubiquitous health data (e.g., containing features of different types and high rate of missing values) and the consumption of various resources (e.g., computing power, time, and cost). Given these two shortcomings, it is dubious whether these studies could translate to realistic settings. In this paper, we propose MuHBoost, a multi-label boosting method for addressing these shortcomings, by leveraging advanced methods in large language model (LLM) prompting and multi-label classification (MLC) to jointly predict multiple health or well-being outcomes. Because LLMs can hallucinate when tasked with answering multiple questions simultaneously, we also develop two variants of MuHBoost that alleviate this issue and thereby enhance its predictive performance. We conduct extensive experiments to evaluate MuH-Boost and its variants on 13 health and well-being prediction tasks defined from four realistic ubiquitous health datasets. Our results show that our three developed methods outperform all considered baselines across three standard MLC metrics, demonstrating their effectiveness while ensuring resource efficiency.

## 1 Introduction

Longitudinal human behavior modeling is an emerging field of study that spans psychology, human-computer interaction, ubiquitous computing, and machine learning (ML). The first three disciplines enable the collection of longitudinal health data (e.g., resting heart rate, sleep quality, and current feelings) from individuals over an extended period of time in their natural environment. Then, ML techniques are utilized to develop predictive models for some relevant health or well-being outcomes (e.g., depression (Chikersal et al., 2021) and atrial fibrillation (Chen et al., 2022)) in the short future. With the advances in mobile and wearable technologies (e.g., smartphones and smartwatches) that facilitate ubiquitous data collection, this multidisciplinary field has widespread implications: from delivering various healthcare services for patients such as chronic disease monitoring, fitness and gait analysis, and obesity management (Saleemi et al., 2023), to developing just-in-time intervention strategies for vulnerable persons who use drugs (PWUDs) (Bae et al., 2018; Tyler et al., 2024).

Data pertaining to longitudinal human behavior modeling, or *ubiquitous health data*[1], contain variables that are collected repeatedly over time and hence are multivariate times series in nature (Xu

---

[*]Correspondence to nate.thach@huskers.unl.edu. [†]University of Nebraska-Lincoln. [§]Lehigh University.

[1]Due to their ubiquitous nature and their intended applications to human health (and well-being).

et al., 2022). Some of these variables are obtained passively (e.g., resting heart rate via smartwatches); others are self-reported by participants (e.g., mood measures via regular smartphone-based questionnaires). The modeling targets for prediction (e.g., depression) can be defined based on either these self-reports or clinical diagnoses from domain experts. Therefore, longitudinal human behavior modeling is often associated with time-series classification (TSC), where the goal is to build models that predict a label for an individual given their longitudinal data across a time period.

While ubiquitous health data are highly informative, they pose unique challenges to standard ML approaches for TSC (Xu et al., 2022). First, they can span a long time period, resulting in long time series. Second, given the *in situ* nature of data collection in order to capture authentic human behaviors, there is a high rate of missing values due to various sources, from participants' low compliance (caused by e.g., privacy concerns or inherently disorganized lifestyle and underlying social pathology) to device failures (e.g., low or dead battery and data transfer loss) (Shiffman, 2009; Intille et al., 2016). Third, they have small sample sizes given that, in order to ensure manageability during data collection, self-reports and clinical diagnoses required to form the labels are administered either infrequently or only at the end of the study (and are occasionally missing as well).

Because of these challenges, Xu et al. (2022) show that traditional ML algorithms (e.g., support vector machine (SVM) (Cortes & Vapnik, 1995), random forest (Ho, 1995), and boosting (Chen & Guestrin, 2016)) and deep learning models (e.g., convolutional neural networks (LeCun et al., 2015) and residual networks (He et al., 2016)) for depression detection struggle to achieve accuracy scores above 50%. (Please refer to the **Related Work** section for further background.) Latest works on longitudinal human behavior modeling (Liu et al., 2023; Englhardt et al., 2024; Kim et al., 2024) attempt to improve the accuracy by employing large language models (LLMs), which demonstrate their strong performance over existing approaches across a wide variety of health prediction tasks at the individual level such as depression and sleep disorder detection with little to no parameter tuning. LLMs also provide a straightforward and effective means to incorporate raw data from different modalities, namely text (from, e.g., domain knowledge and individual auxiliary information), into the model, further improving its predictive performance as shown in all cited works.

**Limitations of Prior Works.** Despite showing promising results, the three aforementioned studies have the two following major shortcomings in terms of practicality. First, they were only validated on health prediction tasks involving numerical time series from mobile and wearable sensing data[2]. This setting is unrealistic given that innovative data collection protocols, such as ecological momentary assessment (EMA) (Shiffman et al., 2008), may result in heterogeneous time series (e.g., categorical and mixed-type data) with even higher rate of missing values. Second, these works do not consider the nontrivial consumption of computing, time, and monetary resources when employing LLMs, which may quickly accumulate given the typical need for predicting multiple related health or well-being outcomes (e.g., comorbidity (Wosiak et al., 2018) and simultaneous usage of multiple drugs (Lorvick et al., 2023)). Please refer to Appendix A for full details.

**Our Contributions.** In this paper, we address these two shortcomings of prior LLM-based methodologies for longitudinal human behavior modeling (i.e., TSC problems concerning ubiquitous health data) by developing a novel method that (i) accommodates more general forms of ubiquitous health data and (ii) minimizes computing, time, and monetary resource consumption without compromising predictive performance. Toward achieving (ii) and motivated by the fact that there often exists some form of interrelationship between different health or well-being outcomes (e.g., anxiety and depression (Kroenke et al., 2009)), we consider the problem of *multi-label classification* (MLC) where multiple tasks associated with these outcomes are simultaneously learned by a shared model. We summarize our contributions as follows:

I. We propose the **M**ulti-**u**biquitous-**H**ealth **Boost**ing (MuHBoost) method that combines state-of-the-art LLM prompting techniques with advanced MLC approaches. In particular, to attain strong predictive performance under various challenges inherent to ubiquitous health data as mentioned earlier, we leverage SummaryBoost (Manikandan et al., 2023), a methodology for integrating a collection of weak learners generated from LLMs into a boosting pipeline. Given its proven promising efficacy in modeling tabular data with high heterogeneity (including missing values) and small sample size, outperforming zero-shot

---

[2]This could be partly attributed to the current lack of public ubiquitous health datasets.

and few-shot approaches by substantial margins without any retraining or finetuning of the employed LLM, we adapt SummaryBoost to (i) work with ubiquitous health data and, more importantly, (ii) enable efficient MLC. By simultaneously inferring multiple health or well-being outcomes, MuHBoost significantly reduces the stated resources needed to employ cutting-edge LLMs for longitudinal human behavior modeling in practice.

II. To mitigate hallucinations from LLMs (Huang et al., 2023) when predicting multiple labels, we develop two variants of MuHBoost by leveraging novel algorithms from MLC literature (Nam et al., 2017; Li et al., 2023). As shown in our experiments, the first variant enhances MuHBoost's predictive performance (see (III) shortly) with simple modifications, and the other generally yields even better performance, albeit with extra costs.

III. We evaluate the effectiveness of MuHBoost and its variants on two publicly available ubiquitous health datasets and two novel datasets concerning the well-being of PWUDs and college students. For each dataset, we define several tasks of interest (2 to 6, for a total of 13 tasks across four datasets) to formulate a single MLC problem. Our results show that MuHBoost and its variants achieve the overall best predictive performance, surpassing all considered baselines, including the three aforementioned state of the art in longitudinal human behavior modeling, across three standard metrics in MLC (Hamming accuracy and micro- and macro-averaged $F_1$). We also provide a time complexity analysis and cost estimation of API calls[7] to demonstrate the resource efficiency of our three developed methods.

**Outline.** In Section 2, we review existing works on longitudinal human behavior modeling and introduce relevant approaches in MLC. In Section 3, we elaborate MuHBoost and its two variants. In Section 4, we evaluate the efficacy of our proposed methods. We conclude our work in Section 5.

## 2 RELATED WORK

Longitudinal human behavior modeling is challenging for existing TSC methods applying to ubiquitous health data due to, e.g., the long time series, the high rate of missing values, the small sample size, and the nature of human behaviors (Xu et al., 2022). Therefore, we first focus on works that directly consider our problem of interest. Then, we discuss MLC approaches upon which MuHBoost and its variants are built. We refer readers to Appendix B for a more comprehensive review.

**Prior Efforts for Longitudinal Human Behavior Modeling.** There has been an increasing interest in building ML models using ubiquitous health data, from detecting mental health disorders (e.g., depression (Burns et al., 2011; Chikersal et al., 2021) and stress (Vos et al., 2023)) to predicting diverse health and well-being outcomes (e.g., atrial fibrillation (Chen et al., 2022), binge drinking (Bae et al., 2018), and glycemic response for dietary recommendations (Zeevi et al., 2015)). However, the generalizability of these models are dubious (Mohr et al., 2017; Saeb et al., 2016; Xu et al., 2022), which prompted the release of several public datasets for the first time, notably GLOBEM (Xu et al., 2022) and LifeSnaps (Yfantidou et al., 2022). Preliminary benchmark results with GLOBEM on a depression detection task (Xu et al., 2022) (same work that released the dataset) show the subpar predictive performance of all 18 existing ML algorithms for the considered problem, calling for methodological advances in longitudinal human behavior modeling. At the same time, research on integrating LLMs in health applications (e.g., generic medical question answering (Singhal et al., 2023) and mental health prediction (Xu et al., 2024)) has flourished over the past years. For this reason, latest works (Liu et al., 2023; Englhardt et al., 2024; Kim et al., 2024) explored the efficacy of LLMs in this area using standard in-context learning (e.g., zero-shot and few-shot prompting) and finetuning techniques, showing they are indeed the state of the art in longitudinal human behavior modeling. Nevertheless, they share the same nontrivial shortcomings as explained in our introduction, which severely undermine their implications for real-world health applications.

**Multi-Label Classification (MLC).** Approaches to MLC are mainly categorized into algorithm adaptation and problem transformation (Gibaja & Ventura, 2015; Herrera et al., 2016). The latter approach, which transforms MLC problems to well-established single-label problems, has seen wider applications. Three most popular transformation methods are binary relevance (BR), classifier chain (CC), and label powerset (LP). BR and CC convert to multiple binary classification problems, whereas LP converts to a multiclass classification problem. BR methods are often employed across

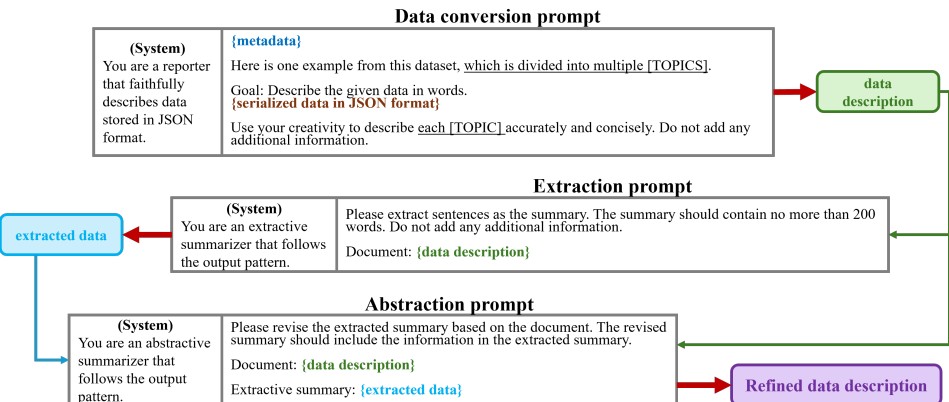

Figure 1: Template for the conversion of a record. Each call to the LLM is denoted by a red arrow.

various fields due to their simplicity and interpretability. However, unlike CC's, they omit the dependencies between labels altogether, which are in fact relevant in health and wellness applications (Wosiak et al., 2018; Blanco et al., 2020; Ge et al., 2020). On the other hand, CC methods are computationally expensive (Read et al., 2021) whereas LP's are prone to suffer from class-wise data scarcity (Tsoumakas & Vlahavas, 2007). Given the discussed underperformance of traditional ML methods and the promise of LLMs in longitudinal human behavior modeling, we approach MLC using ideas from the NLP literature (to be elaborated in Sections 3.2 and 3.3), where the problem is better known as *multi-label text classification* (MLTC) (Chen & Ren, 2021).

## 3    MUHBOOST

We now describe the proposed MuHBoost method. Section 3.1 details our data conversion procedure for adapting SummaryBoost (Manikandan et al., 2023) (originally devised for tabular data) to ubiquitous health data. Section 3.2 elaborates how we construct MuHBoost via extending SummaryBoost to MLC. The two variants of MuHBoost are developed in Section 3.3.

### 3.1    FLEXIBLE DATA CONVERSION

SummaryBoost converts each tabular record into a natural language representation called *data description* by zero-shot prompting the LLM to describe the record as faithfully and concisely as possible. We first devise a similar data conversion procedure for ubiquitous health data, which is highly heterogeneous in practice as mentioned in Section 1. We discuss two main challenges with this adaptation. First, when the record contains either rich contextual information[3] or has long and high-dimensional time series (i.e., large number of features recorded over time), fitting it into the prompt without exceeding the allowed context length of LLMs often turns infeasible or results in a decline in performance (Liu et al., 2024). Second, SummaryBoost's data conversion step does not actively aim to mitigate hallucinations from LLMs[4], which could be a major issue for time series.

To simultaneously address these two concerns, we propose a *two-step* data conversion procedure built on top of the one from SummaryBoost, which is fully automated and requires minimal prompt engineering. As illustrated in Figure 1, the first step (first panel) aims to generate a data description for each record similarly as in SummaryBoost. The metadata contains high-level contexts about the present prediction task, and the serialized data contain individual-level details of a record, which we categorize into *time series* and *auxiliary* data. The latter type of data provides personal contextual information. When data pertaining to a record from either type is rich, we divide it into multiple small chunks or "topics", each of which contains temporally consistent or thematically related in-

---

[3]It has been universally shown that the addition of task-specific (e.g., domain knowledge about the health outcomes to predict) and individual-level (e.g., participants' background) contexts in the prompt can significantly improve the performance of LLMs (Liu et al., 2023; Englhardt et al., 2024; Kim et al., 2024).

[4]With simple directives in the prompt for every API call (e.g., "do not add any additional information").

formation. That is, for time-series data, each topic is a sensible portion of the entire time period e.g., 30-day data comprise "first 10-day", "second 10-day", and "third 10-day" data. For auxiliary data, each topic can cover responses to a block of questions asking for e.g., demographics in some background survey. (Please refer to Appendix C.1 for further details.)

In the second step of data conversion, we combine (if there are multiple topics) and refine the obtained data description(s) of each record into a single final data description. To achieve so, we apply the two-stage pipeline called "extract-then-generate" from Zhang et al. (2023), which was proven to yield significant performance improvement in terms of summary faithfulness (i.e., alleviating hallucination). More specifically, in the first stage (second panel of Figure 1), we prompt the LLM to extract salient information of the converted record. Albeit faithful, *extractive summarization* methods tend to produce rigid and redundant summaries. Therefore, in the second stage of the pipeline (Figure 1's last panel), we prompt the LLM again to revise the extractive summary in order to obtain a more concise and natural-sounding data description, which is known as *abstractive summarization* (Gupta & Gupta, 2019). Finally, in accordance with SummaryBoost, we append the associated label information at the end of each refined data description. In the case of binary MLC (i.e., all labels are binary), to reduce hallucinations from negation (Truong et al., 2023) in records having mostly negative labels (e.g., from healthy participants), we only concatenate a binary label of the corresponding multi-label record if it is positive[5]. If a record does not have any positive label, we instead append an explicitly negative sentence such as "Hence, this participant does not have any [disorders]."

## 3.2 EFFICIENT MULTI-LABEL BOOSTING

At its core, SummaryBoost leverages LLMs to generate summaries from a representative subset of the dataset (given the limited context length of LLMs). These summaries then serve as weak learners in a traditional boosting algorithm, particularly AdaBoost (Freund & Schapire, 1997), for binary or multiclass classification. This concept can be neatly extended to MLC via the LP approach.

Recall that LP reduces a $Q$-label classification problem to a multiclass classification problem, where each class corresponds to one of the $2^Q$ unique label combinations. Notice that the actual number of classes is $\min\{N, 2^Q\}$, where $N$ is the sample size. Motivated by (Song et al., 2023), which converts MLTC into a prompt learning task, we can design a generalizable prompt template in order to help LLMs capture the potential semantic relationships between multiclass labels and text from a given dataset. Concretely, given a set of data descriptions, we first ask the LLM to generate multiple (i.e., $Q$) summaries, one specifically for each label, then concatenate them into a multi-label weak learner, which is later fed into the LLM for multiclass classification. The remainder of SummaryBoost algorithm is kept mostly intact with two minor changes. First, we replace the simple error loss in binary classification with *subset 0/1* loss for binary MLC problems (Nam et al., 2017). We then set $K$, the number of classes (needed for computing the early stopping threshold and the coefficient of the weak learner at each boosting round), to $\min\{N, 2^Q\}$.

Although the present extension is sufficient for $2^Q \ll N$, this assumption can be invalid given the generally small sample size of ubiquitous health datasets. In other words, it is possible to encounter situations where $2^Q$ approaches $N$ i.e., each sample belongs to its own class. This trivializes the originally proposed `ClusterSampling` subroutine, which selects a representative subset of the training data for each class such that same-class samples from smaller clusters (determined using *hierarchical agglomerative clustering* (Nielsen, 2016) on the language embeddings of each data description in SummaryBoost) are more likely to be chosen. That is, when there are close to $N$ different classes, clustering is clearly unnecessary because there is only one or two samples to consider, and hence `ClusterSampling` would return more or less the same set of input samples, which is meaningless as its purpose is to select $s < N$ representative samples from the training set. Thus, we modify this subroutine as shown in Algorithm 1. Conceptually, if the size of the representative subset is not substantially larger than the number of possible classes (Step 2), then we simply sample based on how "informative" the records are (i.e., having as many positive labels that are as rare as possible) without having to do `ClusterSampling`. Otherwise, we can also skip clustering for classes with no more than two examples (Step 13). Since this addition could return more than $s$ samples, we consider rarer classes $k$ first in the for loop at Step 11. Moreover, if for some $k$, $|S| + (s \times r[k]) > s$, we only sample $s - |S|$ examples in Step 16 and return $S$ preemptively.

---

[5]The order of the labels to concatenate will be discussed shortly.

---

**Algorithm 1 ClusterSampling** (Extended)

---

**Input:** Training data $\mathbf{X}_{train}$ with sample size $N$, label $\mathbf{y}_{train} = \{y_i\}_{i=1}^N$ where $y_i \in \{0,1\}^Q$, sample distribution $\mathbf{w}$ for the current boosting round, size of the representative subset $s$, array of class ratios $r$, array $v$ of length $Q$, positive integer $\kappa$.

1: $S \leftarrow$ new empty set          ▶ $r[k]$ and $v[q]$ are the proportion of samples in class $k$ and of positive samples for label $q$, respectively.
2: **if** $s < \min\{N, 2^Q\} + \kappa$ **then**
3:     $\boldsymbol{w} \leftarrow$ new array of same length as $\mathbf{w}$ filled with 1
4:     **for** $i = 1$ to $N$ **do**
5:        **for** $q = 1$ to $Q$ such that $y_i^q = 1$ **do**
6:           $\boldsymbol{w}[i] \leftarrow \boldsymbol{w}[i] \times \frac{1}{v[q]}$
7:     $\boldsymbol{w} \leftarrow \texttt{Normalize}(\texttt{Normalize}(\boldsymbol{w}) \odot \mathbf{w})$      ▶ re-normalizes the sample weights; $\odot$ denotes element-wise multiplication
8:     Sample $s$ examples from $\mathbf{X}_{train}$ using $\boldsymbol{w}$ and append to $S$      ▶ sample *without* replacement (same applies to Step 16)
9: **else**
                                                      ▶ only do clustering if $s \gg 2^Q$
10:     $\boldsymbol{w} \leftarrow$ new array of same length as $\mathbf{w}$ filled with -1 (for initialization)
11:     **for** $k = 1$ to $\min\{N, 2^Q\}$ **do**
12:        $\mathbf{X}_{train}^k \leftarrow \{\mathbf{X}_{train}[i]\}_{i=1}^{r[k] \cdot N}$      ▶ samples that belong to class $k$
13:        **if** $|\mathbf{X}_{train}^k| \leq 2$ **then**
14:           Append them to $S$ and **continue** for next $k$
15:        Obtain $\boldsymbol{w}$ based on (vanilla) $\texttt{ClusterSampling}(\mathbf{X}_{train}, \mathbf{y}_{train}, r, \mathbf{w}, s)$      ▶ lines 5-14 of Algorithm 1
16:        Sample $s \times r[k]$ examples from $\mathbf{X}_{train}$ using $\boldsymbol{w}$ and append to $S$      in (Manikandan et al., 2023)
**Return:** $S$      ▶ used as input to the LLM for generating summaries

---

Hence, our modification also helps deal with the rare label issue where a label may have imbalanced distribution of binary classes. We refer to our approach discussed so far as **MuHBoost**.

### 3.3 ENHANCING MuHBOOST

We now describe two variants of MuHBoost based on advanced LP and CC methods as follows.

**MuHBoost[LP+].** One potential issue with MuHBoost is for larger $Q$, the increased computational burden due to longer and more complex prompts may lead to hallucinations from LLMs (Huang et al., 2023; Liu et al., 2024) and hence worsen predictive performances. In response, we propose a simple-yet-effective modification inspired by (Nam et al., 2017) in MLTC, which reformulates LP into the problem of identifying all $Q' \leq Q$ *positive* labels of a given $Q$-label sample. Essentially, instead of inferring the class $k \in \{1, \ldots, \min\{N, 2^Q\}\}$

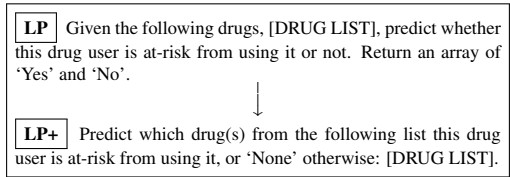

Figure 2: Template for inference directives of MuHBoost and MuHBoost[LP+] for our local dataset concerning PWUDs.

that corresponds to the simultaneous prediction of all $Q$ labels for a sample, we predict which of its labels are positive in a sequential manner. While this idea was initially proposed on traditional language models (RNN, GRU, and Seq2Seq) and hence required extensive architecture design, we can achieve this with LLMs by redesigning the inference directive in the inference prompt such that instead of asking for binary answers (e.g., an array of 'Yes' or 'No'), we prompt the names of the positive labels, if any, as shown in Figure 2. We refer to this approach as **MuHBoost[LP+]**.

**MuHBoost[CC].** Our second MuHBoost variant involves CC, with motivations and design objectives elaborated in Appendix D.2. Given that SummaryBoost utilizes AdaBoost as its backbone, we focus on latest works in MLC that combine AdaBoost with CC. In particular, we leverage AdaBoost.C2 (Li et al., 2023), a multi-path AdaBoost framework where a boosting path is established independently for each label, each of which aims to minimize the decomposed loss of the corresponding label while still being linked together in a chain. This allows flexibility during training since each of the $Q$ boosting processes has its own set of weights and is hence able to fit the respective label to the fullest extent as well as to early-stop asynchronously when needed. As a result, the label-wise predictive performance is maximized and the time complexity is effectively reduced. Our adaptation of AdaBoost.C2 to MuHBoost, called **MuHBoost[CC]**, is formalized in Algorithm 2. Note that since we mainly consider binary MLC[6], we set the number of classes $K$ to 2 beforehand.

---

[6]Generalizing MuHBoost[CC] to *multiclass* MLC (i.e., some or all of the $Q$ labels are multiclass) is straightforward and we defer its employment to future work.

---

**Algorithm 2 MuHBoost[CC]**

---

**Input:** Training data $\mathbf{X}_{train}$ with sample size $N$, label matrix $\mathbf{Y}_{train}$ of size $(N \times Q)$, maximum number of boosting rounds $T$, size of the representative subset $s$, permutation function $\sigma$, hyperparameters $\delta > 0$, $\mu \in [0, 0.5)$.

1: $\mathbf{H}, \mathbf{A}, \mathbf{O} \leftarrow$ new empty matrices of size $(T \times Q)$
2: $\mathbf{W} \leftarrow$ new matrix of size $(Q \times N)$ filled with $\frac{1}{N}$         ▶ $\mathbf{W}[q, :]$ is the sample distribution for label $q$
3: $o_t \leftarrow \sigma(\{1, \ldots, Q\})$        ▶ $o_t$ is the ordering of labels at round $t$ containing at most $Q$ elements
4: **for** $t = 1$ to $T$ **do**
5:    $\mathbf{O}[t, :] \leftarrow o_t;$    $Q' \leftarrow |o_t|;$    $\epsilon \leftarrow$ new array of length $Q'$
6:    $\mathbf{X}'_{train} \leftarrow \mathbf{X}_{train} \oplus \mathbf{Y}_{train}[:, q']$ for $q' \in \{1, \ldots, Q\} \setminus o_t$
7:    **for** $q = 1$ to $Q'$ **do**
8:      $(\mathbf{X}_s, \mathbf{Y}_s) \leftarrow \texttt{ClusterSampling}(\mathbf{X}'_{train}, \mathbf{Y}_{train}[:, q], \mathbf{W}[q, :], s)$   ▶ Algorithm 1 in (Manikandan et al., 2023)
9:      $\mathbf{H}[t, q] \leftarrow \texttt{Summary}(\mathbf{X}_s, \mathbf{Y}_s)$     ▶ Summary prompts the LLM to summarize the representative samples
10:      $\epsilon[q] \leftarrow \frac{\sum_{i=1}^{N} \mathbf{W}[q,i] \times \mathbb{1}\{\mathbf{H}[t,q](\mathbf{X}_{train}[i]) \neq \mathbf{Y}[i,q]\}}{\sum_{i=1}^{N} \mathbf{W}[q,i]}$    ▶ $\epsilon[q]$ is the training error rate for label $q$
11:      **if** $\epsilon[q] \geq 0.5 - \mu$ **then**     ▶ $\mu \geq 0$ enforces higher-quality summaries for accelerating convergence
12:        Go to Step 8
13:      $\mathbf{A}[t, q] \leftarrow \log\left(\frac{1-\epsilon[q]}{\epsilon[q]}\right)$ if $\epsilon[q] \geq \delta$ else $0$
14:      **if** $\epsilon[q] \geq \delta$ **then**
15:        **for** $i = 1$ to $N$ **do:**    $\mathbf{W}[q, i] \leftarrow \mathbf{W}[q, i] \times e^{\mathbf{A}[t,q] \times \mathbb{1}\{\mathbf{H}[t,q](\mathbf{X}_{train}[i]) \neq \mathbf{Y}[i,q]\}}$
16:        $\mathbf{W}[q, :] \leftarrow \texttt{Normalize}(\mathbf{W}[q, :])$
17:      $\mathbf{X}'_{train} \leftarrow \mathbf{X}'_{train} \oplus \mathbf{Y}_{train}[:, q]$   ▶ $\oplus$ denotes the appending of true label(s) (if positive only) to each training sample
18:      **if** $\epsilon[q] < \delta$ **then**
19:        Remove $q$ from $o_t$    ▶ for sufficient $t < T$, the rows after the $t$th row of $\mathbf{H}, \mathbf{A}, \mathbf{O}$ can have less than $Q$ elements
**Return:** $\mathbf{H}, \mathbf{A}, \mathbf{O}$        ▶ used during inference of unseen samples (see Equations S1 and S2)

---

For determining the label ordering, we opt for a reasonable chain order defined by some heuristic permutation $\sigma$ and fix it throughout the boosting process. Please refer to Appendix D.2 for an in-depth explanation. Note that in addition to chain ordering, we also order the labels according to $\sigma$ for the label information concatenated to each data description as well as the e.g., [DRUG LIST] in the summary and inference directives (Figure 2). The same applies to MuHBoost and MuHBoost[LP+].

## 4   EXPERIMENTS

We start this section by describing the considered ubiquitous health datasets and their associated tasks in Section 4.1. In Section 4.2, we detail the experimental setups for validating the efficacy of MuHBoost and its variants, including the relevant baselines and our evaluation criteria. Finally, we present our results in Section 4.3. Appendix C.2 includes further implementation details.

### 4.1   DATASETS AND TASKS

We consider four ubiquitous health datasets that contain both time-series and auxiliary data as well as sufficient information to form multiple labels for each: LifeSnaps (Yfantidou et al., 2022), GLOBEM (Xu et al., 2022), college students (CoSt) (Hayat & Hasan, 2023), and PWUD (Tyler et al., 2024). The first two are publicly available and have previously been considered by Englhardt et al. (2024); Kim et al. (2024) (as mentioned in Section 2), whereas the latter two are novel and require submitting an IRB protocol and ethical research plan to their authors. For each dataset, let $D_{ts}$ and $D_{aux}$ be the dimensionality of time series and auxiliary data, respectively. We then formulate an MLC problem by defining $Q \in [2, 6]$ prediction tasks that consider the same set of features, with or without the auxiliary data. Please refer to Appendix C.1 for the exact features used in each task.

**LifeSnaps** ($N = 130$)**.** We consider two tasks ($Q = 2$) predicting whether each sample would be at risk from: (i) *negative emotions*, defined by having PANAS negative affect score (Watson et al., 1988) higher than positive affect score, and (ii) *anxiety*, defined by having above average STAI stress score (Spielberger et al., 1999), given daily time-series data ($D_{ts} = 8$) in the past seven days in accordance with (Kim et al., 2024). The associated auxiliary data ($D_{aux} = 28$) are extracted from the health and psychological baseline surveys pertaining to the participants upon entry.

**GLOBEM** ($N = 96$)**.** We consider three tasks ($Q = 3$) predicting whether each sample would be at risk from: (i) *negative emotions*, defined by having a PANAS negative affect score of 6 or greater, (ii) *anxiety and depression*, defined by having total PHQ-4 score (Kroenke et al., 2009) greater than 5 (otherwise none if score lower than 1), and (iii) *stress*, defined by having a total PSS-4 score (Cohen et al., 1983) of 6 or greater. Since the three surveys for forming these labels were not all

delivered within the same day, while we still keep the size of each sample as $28(\text{days}) \times 16(D_{ts})$ in accordance with (Englhardt et al., 2024), we define its corresponding (multi-)label with respect to the scores from these three surveys within 3–4 days after the 28th day. Following (Englhardt et al., 2024), we then create a balanced dataset by randomly sampling 32 data points from each year (Years 2, 3, 4) with a uniform distribution of labels (4 for each of the $2^3 = 8$ label combinations), for a total of 96 samples. The auxiliary data ($D_{aux} = 27$) are derived from the "pre surveys", which contain questionnaires asking for personality, physical health, mental well-being, and social well-being.

**CoSt** ($N = 48$). This dataset was obtained from 48 college students enrolled in an introductory programming course at the University of Nebraska-Lincoln, in which their self-evaluations on course performance and motivation (intrinsic and extrinsic) factors throughout the 16-week semester were recorded weekly via the aforementioned ODIN app (Khan et al., 2019). We consider two tasks ($Q = 2$) predicting whether each student would be at risk of (i) underperforming in Project 1 (constitutes 10% of the final grade), defined by receiving a grade below 8 (out of 10) and (ii) failing the course, defined by receiving a final letter grade of C+ or lower, given the stated weekly time-series data ($D_{ts} = 11$) from the first eight weeks. We formed these definitions with the goal of ensuring there are sufficient positive samples for each task. The auxiliary data ($D_{aux} = 26$) comprise their background information (e.g., declared major, parents' highest level of education, total household income, etc.) acquired from an intake survey as well as grades of various lesser course items (e.g., labs, quizzes, homework assignments, etc.) that were due within the first eight weeks.

**PWUD** ($N = 68$). This dataset consists of 100 PWUDs from Nebraska, U.S. whose substance use, sociality, stress, and support were enquired daily via the ODIN app over the span of 30 days, with $D_{ts} = 22$. PWUDs also took an intake and an exit survey (same for both) about drug uses and individual attributes at the start and the end of the study, respectively. We construct auxiliary data ($D_{aux} = 108$) using the intake survey and create labels using the exit survey. Hence, we require the period between the two taken surveys to be close to 30 days (i.e., within 27–33 days), and the period from the last answer in the ODIN app to the exit survey completion to be be small (i.e., within 3 days), which results in $N = 68$. Our six considered tasks ($Q = 6$) involve predicting whether PWUDs would be at risk of using: (i) alcohol, (ii) marijuana, (iii) methamphetamine, (iv) cocaine, (v) amphetamines, and (vi) opioids. After conferring with domain experts in substance use research, we define: (i) as having binge drinking; (ii) as using more than once a week; (iii) and (iv) as using once a month or more; and (v) and (vi) as using once a week or more.

## 4.2 Experimental Setups

**Baselines.** We consider the following methods from both problem transformation (•) and algorithm adaptation (∗) approaches for MLC. We refer readers to Appendix C.2 for full details.

(•) *Zero-shot*: Similarly as in the work of SummaryBoost, we query the LLM with a data description and directly infer its label. For MLC, this can be done in three ways: asking for either (i) $Q$ labels one at a time (hence $Q$ separate calls to the LLM), (ii) $Q$ labels simultaneously, or (iii) only its positive labels, which are reminiscent of BR, LP, and LP+ approaches, respectively. We therefore denote them as **0-shot[BR]**, **0-shot[LP]**, and **0-shot[LP+]**. Note that for single-label problems ($Q = 1$), 0-shot[BR] is equivalent to the zero-shot prompting techniques developed in (Englhardt et al., 2024; Kim et al., 2024). We use GPT-4 (`gpt-4-turbo`) following (Kim et al., 2024).

(•) *Few-shot*: We follow the few-shot setting as described in SummaryBoost's experiments, where we leverage their `ClusterSampling` to select a representative subset of $\mathbf{X}_{train}$ for demonstration before asking the LLM to infer the label of some unseen sample. We set the number of shots to 10, which is consistent with $s$ fixed earlier. Similar to the zero-shot setting, there are three ways to ask the LLM for MLC, for which we denote as **10-shot[BR]**, **10-shot[LP]**, and **10-shot[LP+]**. We also use GPT-4 for the same stated reason.[12]

(•) *Random forest* (Ho, 1995) and *XGBoost* (Chen & Guestrin, 2016): We consider problem transformation via both CC and LP when employing these baselines for MLC, for which we denote respectively as **RF[CC]**, **RF[LP]**, **XGBoost[CC]**, and **XGBoost[LP]**.

(∗) *Multi-label $k$ Nearest Neighbours* (**ML$k$NN**) (Zhang & Zhou, 2007) and *twin multi-label SVM* (**MLTSVM**) (Chen et al., 2016): Lastly, we include two well-established methods for MLC via algorithm adaptation, which respectively adapts $k$NN and SVM to MLC directly.

| Method | LifeSnaps | LifeSnaps+ | GLOBEM | GLOBEM+ | CoSt | CoSt+ | PWUD | PWUD+ |
|---|---|---|---|---|---|---|---|---|
| 0-shot[BR] | [16 16 17] | [14 15 14] | [18 17 16] | [13 15 15] | [14 14 13] | [13 12 12] | [14 14 15] | [13 12 13] |
| 0-shot[LP] | [18 18 19] | [15 14 15] | [17 18 18] | [15 16 14] | [17 18 18] | [15 16 16] | [16 18 19] | [17 16 17] |
| 0-shot[LP+] | [17 17 16] | [12 13 13] | [16 14 17] | [10 12 11] | [18 17 17] | [16 15 15] | [18 17 16] | [15 15 14] |
| 10-shot[BR] | [13 12 11] | [ 8  7  7] | [12 10 10] | [ 8  9  7] | [10  9  9] | [ 7  8  7] | [ 8 10  8] | [ 7  7  6] |
| 10-shot[LP] | [11 10 12] | [ 9  9  8] | [11 11 13] | [ 9  8  9] | [12 13 14] | [ 9 11 10] | [12 13 12] | [ 9 11 11] |
| 10-shot[LP+] | [10 11 10] | [ 7  8  9] | [14 13 12] | [ 7  7  8] | [11 10 11] | [ 8  6  8] | [11  9 10] | [10  8  9] |
| RF[CC] | [21 19 18] | [22 21 20] | [23 22 19] | [24 23 23] | [30 27 29] | [27 29 30] | [28 30 30] | [26 29 28] |
| RF[LP] | [28 29 30] | [29 27 29] | [27 26 27] | [28 29 30] | [25 24 22] | [24 23 25] | [25 19 20] | [22 24 22] |
| XGBoost[CC] | [19 20 23] | [20 22 21] | [22 21 20] | [20 19 24] | [23 25 27] | [26 28 24] | [24 23 27] | [27 28 24] |
| XGBoost[LP] | [25 30 28] | [30 26 25] | [21 24 21] | [29 28 26] | [28 30 28] | [29 26 26] | [29 27 29] | [30 26 25] |
| ML$k$NN | [24 23 22] | [26 24 24] | [19 20 22] | [30 25 28] | [21 19 20] | [19 22 23] | [23 20 18] | [21 21 23] |
| MLTSVM | [23 25 27] | [27 28 26] | [25 27 29] | [26 30 25] | [22 20 21] | [20 21 19] | [19 25 21] | [20 22 26] |
| MuHBoost | [ 6  6  5] | [ 2  2  3] | [ 5  6  6] | [ 3  3  3] | [ 6  7  6] | [ 4  5  5] | [ 6  6  7] | [ 5  5  6] |
| MuHBoost[LP+] | [ 5  4  6] | [ 3  3  2] | [ 6  4  4] | [ **1**  2  2] | [ 5  4  4] | [ 2  2  2] | [ 3  3  4] | [ **1**  **1**  **1**] |
| MuHBoost[CC] | [ 4  5  4] | [ **1**  **1**  **1**] | [ 4  5  5] | [ 2  **1**  **1**] | [ 3  3  3] | [ **1**  **1**  **1**] | [ 4  4  3] | [ 2  2  2] |

Table 1: Performance ranking ($\downarrow$) of 15 methods subject to [HA mi$F_1$ ma$F_1$]. (+) next to the datasets' tags denotes incorporation of auxiliary data in addition to time-series data (hence there are $15 \times 2 = 30$ methods in total).

**Evaluation Metrics and Procedure.** For each considered set of experimental configurations, we used the split ratio of 50/10/40 (for train/validation/test set), which follows SummaryBoost's evaluation, for a total of 10 different splits. For partitioning multi-label datasets into training (train+validation) and test sets, we adopt the *iterative stratification* algorithm[13] (Sechidis et al., 2011). We use the standard *Hamming accuracy* (HA) $= \frac{1}{Q} \sum \mathbb{1}\{\hat{\boldsymbol{y}} = \boldsymbol{y}\} \in [0, 1]$ metric for MLC (Tsoumakas & Katakis, 2007), where $\boldsymbol{y}, \hat{\boldsymbol{y}}$ are vectors of ground-truth labels of a given sample and their corresponding predictions, and report the resulting summed-then-averaged value from all test samples. Since we encounter class imbalance for individual labels in 3 out of 4 datasets, we also consider *micro-* and *macro-averaged-$F_1$-measure* (Herrera et al., 2016). They are respectively defined by Nam et al. (2017) as mi$F_1 = \frac{\sum_{q=1}^{Q} 2\mathrm{TP}_q}{\sum_{q=1}^{Q} (2\mathrm{TP}_q + \mathrm{FP}_q + \mathrm{FN}_q)}$ and ma$F_1 = \frac{1}{Q} \sum_{q=1}^{Q} \frac{2\mathrm{TP}_q}{2\mathrm{TP}_q + \mathrm{FP}_q + \mathrm{FN}_q}$, where TP, FP, FN stand for true positives, false positives, and false negatives from all test samples. Whereas mi$F_1$ evaluates the overall performance of a classifier (and hence rewards those giving correct predictions on frequent labels), ma$F_1$ focuses on assessing performance on rare labels.

## 4.3 Results

Table 1 shows the predictive performance of our methods relative to the considered baselines, where each bracket contains rankings of the respective method ($\in \{1, \ldots, 30\}$, including whether auxiliary data is incorporated or not) for the three aforementioned MLC metrics. MuHBoost, with or without integrating auxiliary data into the models, already outperforms all baselines in most cases. Its variants, [LP+] and [CC] for brevity, bring further improvements and hence yield the overall best performance. For MLC problems with small $Q$ (from LifeSnaps, GLOBEM, and CoSt), [CC] seems to perform better than [LP+]; whereas the contrary holds for the remaining MLC problem with larger $Q$ (from PWUD dataset). As expected, traditional supervised ML methods perform poorly across all datasets, trailing few-shot and even zero-shot methods. For those leveraging LLMs including ours, we also see that incorporating auxiliary data in the prompt brings enhanced performance. This observation agrees with prior works on modeling ubiquitous health data (Liu et al., 2023; Englhardt et al., 2024; Kim et al., 2024), which demonstrated the significance of providing rich contextual information when asking LLMs for inference. The same, however, does not apply for methods from classical ML algorithms, which could be due to overfitting given their lack of encoded prior knowledge as well as the small sample size and high dimensionality of all employed datasets.

**Impact of Refining Data Descriptions.** We investigate the effect on predictive performance of the refinement step (i.e., abstractive summarization in the bottom panel of Figure 1) in our proposed data conversion procedure. (We omitted vanilla MuHBoost due to its lesser performance compared to [LP+] and [CC]; and we did not ablate the extractive summarization step since it is required to trim each data description within a reasonable word limit.) Recall that this step enforces conciseness and fluidity in the data descriptions. As shown in Table 2 (where $b$ from $a(b)$ is the standard deviation expressed in terms of the two least significant digits in $a$ (Bipm et al., 1995)), our ablation

demonstrates its positive impact across all datasets, which is reasonable since such refinement of information effectively alleviates hallucinations from the LLM.

| Dataset | MuHBoost[LP+] | | MuHBoost[CC] | |
|---|---|---|---|---|
| | **Refine data descriptions?** | | | |
| | No | Yes | No | Yes |
| LifeSnaps+ | 0.704(42) | 0.731(29) | 0.714(53) | **0.739(47)** |
| | 0.798(65) | 0.836(46) | 0.806(59) | **0.843(56)** |
| | 0.779(71) | 0.822(40) | 0.787(64) | **0.830(58)** |
| GLOBEM+ | 0.676(54) | **0.720(43)** | 0.692(49) | 0.718(38) |
| | 0.795(66) | 0.829(62) | 0.807(57) | **0.833(47)** |
| | 0.784(70) | 0.817(59) | 0.796(59) | **0.825(51)** |
| CoSt+ | 0.697(45) | 0.715(38) | 0.699(44) | **0.726(52)** |
| | 0.782(60) | 0.814(54) | 0.777(47) | **0.823(59)** |
| | 0.770(58) | 0.803(49) | 0.764(51) | **0.812(48)** |
| PWUD+ | 0.659(63) | **0.688(54)** | 0.662(47) | 0.673(39) |
| | 0.746(59) | **0.784(48)** | 0.739(55) | 0.775(44) |
| | 0.727(52) | **0.771(41)** | 0.721(49) | 0.769(36) |

Table 2: Ablation of the abstractive summarization step in our data conversion procedure. In each cell: (top) HA, (middle) $miF_1$, (bottom) $maF_1$.

| Dataset | MuHBoost[LP+] | | MuHBoost[CC] |
|---|---|---|---|
| | GPT-3.5 | **GPT-4** | GPT-3.5 |
| LifeSnaps+ | 0.731(29) | 0.733(32) | **0.739(47)** |
| | 0.836(46) | 0.839(50) | **0.843(56)** |
| | 0.822(40) | 0.825(47) | **0.830(58)** |
| GLOBEM+ | **0.720(43)** | 0.711(47) | 0.718(38) |
| | 0.829(62) | 0.813(58) | **0.833(47)** |
| | 0.817(59) | 0.806(55) | **0.825(51)** |
| CoSt+ | 0.715(38) | 0.718(42) | **0.726(52)** |
| | 0.814(54) | 0.820(45) | **0.823(59)** |
| | 0.803(49) | **0.815(52)** | 0.812(48) |
| PWUD+ | 0.688(54) | **0.696(53)** | 0.673(39) |
| | 0.784(48) | **0.789(62)** | 0.775(44) |
| | 0.771(41) | **0.781(61)** | 0.769(36) |

Table 3: Ablation of the underlying GPT model in MuHBoost[LP+]. In each cell: (top) HA, (middle) $miF_1$, (bottom) $maF_1$.

**Upgrading to GPT-4.** Table 3 shows the effect of switching to GPT-4 for MuHBoost[LP+]. (Doing the same for [CC] costs $\times Q$ more, which is prohibitively expensive given the substantially higher cost of calling GPT-4 compared to GPT-3.5 as specified earlier.) Although leveraging GPT-4 helps slightly improve the performance of [LP+], it does not bring consistent improvement over [CC], especially in small $Q$ settings (from the first three datasets). Even when $Q$ is larger (from PWUD dataset), the improvement with respect to those utilizing GPT-3.5 is small.

**Discussion (more in Appendix C.3). Resource Consumption:** MuHBoost and its variants require minimal computing power by leveraging commercial LLMs. In terms of runtime, MuHBoost and [LP+] share the complexity with SummaryBoost and therefore take $O(TMf)$, where $T$ is the number of boosting rounds, $M$ is the number of resampling for the weak learner at each round, and $f$ is the LLM's runtime. In terms of cost associated with training a model via these two methods, according to Manikandan et al. (2023), the upper bound for the total number of requested tokens is $T \times [M \times (S_t + 0.5N \times P_t) + 0.1N \times P_t]$, where $S_t$ and $P_t$ are the context length of the LLMs employed for summarization and for inference (same in our experiments), respectively. For [CC], an extra factor $Q' \leq Q$ is added for both time and cost. For a comparison with the state of the art in longitudinal human behavior modeling, the finetuning approach from (Kim et al., 2024) (exhibiting the overall best performance) takes $O(QTNf)$ in total for an MLC problem. Thus, our developed methods offer a resource-efficient as well as effective solution for modeling ubiquitous health data. **[LP+] versus [CC]:** When comparing between the two MuHBoost variants that yield the overall best predictive performance, [CC] seems to excel for small to moderate $Q$ and when there are less constraints on time and budget, whereas [LP+] is applicable to all other cases (see Figure S3). **Interpretability:** Following SummaryBoost, MuHBoost and its variants also provide interpretability by storing LLM-generated knowledge regarding the task at hand into natural-language summaries.

## 5 CONCLUSION

In this paper, we propose MuHBoost, a multi-label boosting method for longitudinal human behavior modeling, along with its two variants for addressing two practical aspects that are lacking in prior works: the heterogeneity in real-world ubiquitous health data and the consumption of computing, time, and monetary resources. We achieve so by adapting SummaryBoost, a state-of-the-art method for heterogeneous and small tabular data prediction using LLMs, to our problem of interest, and then extending it to multi-label classification settings where multiple related tasks can be simultaneously learned. Our experiments covering 13 health and well-being prediction tasks from four diverse datasets demonstrate the effectiveness of all developed methods, in addition to their resource efficiency as well as interpretability. Our work can thus potentially help domain experts from various medical and social science disciplines better understand human behaviors and develop effective personalized prevention or intervention strategies for at-risk individuals accordingly.

ACKNOWLEDGMENTS

We would like to thank all ICLR-25 reviewers for the valuable feedback and the interactive discussion during the rebuttal period, which significantly improve the overall quality of our work. Our project is supported by the National Institute of General Medical Sciences of the National Institutes of Health [P20GM130461], the Rural Drug Addiction Research Center at the University of Nebraska-Lincoln, and the National Science Foundation under grant IIS:RI #2302999 and IIS:RI #2414554. The content is solely the responsibility of the authors and does not necessarily represent the official views of the funding agencies.

ETHICS STATEMENT

Our work proposes a novel method powered by large language models. Although achieving great promises, there could be potential biases stemming from subjective, biased, and ungrounded text corpora as well as quality degradation over time from these language models (e.g., ChatGPT (Chen et al., 2024)). Therefore, we do not advise on the use of the proposed method for determining at-risk individuals on the fly, but rather for forming a decision-aid system that complements decision-making from human experts.

REPRODUCIBILITY STATEMENT

We refer readers to Sections 4.1 and 4.2 as well as Appendices C.1 and C.2 for complete details on reproducing our results, including the link to our anonymous GitHub repository[10].

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

## A    LIMITATIONS OF PRIOR WORKS

Despite showing promising results with LLMs, the studies conducted by Liu et al. (2023); Englhardt et al. (2024); Kim et al. (2024) have two following major shortcomings in terms of practicality.

- They were only validated on health prediction tasks involving numerical time series from mobile and wearable sensing data[2]. This setting is unrealistic given that innovative data collection protocols, such as ecological momentary assessment (EMA) (Shiffman et al., 2008), enable the acquisition of fine-grained behavioral data not just passively via sensors, but also actively via e.g., smartphone-based personalized surveys asking participants multiple-choice questions routinely. Hence, there could be non-numerical time series such as categorical (from either single- or multi-select questions) and those with mixed data types (from questions that ask for either e.g., a free-form input or "not applicable"). Furthermore, advanced technologies (e.g., SARA (Rabbi et al., 2018) and ODIN (Khan et al., 2019)) that facilitate *in situ* data collection could bring further heterogeneity in the resulting dataset by different means. The SARA mobile application, for example, provides various engagement strategies for incentivizing participants to self-report on their day (e.g., a virtual aquarium in the app grows with fish as they complete more daily self-reports). It has been shown in practice, however, that such intricate data collection platform could lead to technical software issues where survey data from self-reports are not consistently saved Coughlin et al. (2021). In another example, the ODIN software platform for implementing EMA methods offers specification of contextual rules that govern when each question will be asked (e.g., at certain times of the day, geographic locations, or even on button press from a selection to some currently prompted question). Although this functionality could provide an effective means to gather contextual data, it may complicate data analytics in return. Most evidently, when a question is omitted by design, the current entry for the corresponding time series is recorded as a missing value. As a result, in both examples, the already high missing rate from ubiquitous health data is inadvertently exacerbated.
- There were no considerations on resource consumption when employing the LLMs via finetuning as well as zero-shot and few-shot prompting approaches. Finetuning even small LLMs as in (Kim et al., 2024) requires significant amounts of computing power and runtime, whereas zero-shot and few-shot techniques rely on much bigger LLMs that must be pretrained on enormous text corpora to be effective, as implemented in (Liu et al., 2023). On the other hand, utilizing commercial LLMs via their APIs as in (Englhardt et al., 2024; Kim et al., 2024) requires paying a nontrivial fee[7]. Furthermore, such consumption of com-

---

[7]Each API call to commercial LLMs (e.g., OpenAI's GPT family, Meta's Llama, and Google's Gemini) incurs a specific dollar cost subject to the number of input/output tokens.

puting, time, and monetary resources may quickly accumulate given the typical need for predicting multiple related health or well-being outcomes (e.g., comorbidity (Wosiak et al., 2018) and simultaneous usage of multiple drugs (Lorvick et al., 2023)). In fact, Liu et al. (2023); Kim et al. (2024) defined multiple tasks (up to 5) associated with each considered dataset, such as anxiety and depression detection with GLOBEM (Kim et al., 2024) (where they built a separate model for each).

# B  RELATED WORK (LONG VERSION)

**Prior Efforts for Longitudinal Human Behavior Modeling.**  There has been an increasing interest in building ML models using ubiquitous health data, from detecting mental health disorders (e.g., depression (Burns et al., 2011; Chikersal et al., 2021) and stress (Vos et al., 2023)) to predicting diverse health and well-being outcomes (e.g., atrial fibrillation (Chen et al., 2022), binge drinking (Bae et al., 2018), and glycemic response for dietary recommendations (Zeevi et al., 2015)). However, these models either are inaccurate or require hand-crafted features to be effective, and every one of them was validated on a single, private dataset (Mohr et al., 2017; Xu et al., 2022). Furthermore, Saeb et al. (2016) found that a substantial portion of earlier works used inappropriate cross-validation techniques, resulting in overestimated predictive performances. To overcome these generalizability concerns, several datasets have been publicly released for the first time, notably GLOBEM (Xu et al., 2022) and LifeSnaps (Yfantidou et al., 2022). Preliminary benchmark results with GLOBEM on a depression detection task (Xu et al., 2022) (same work that released the dataset) show the subpar predictive performance of all 18 existing ML algorithms for the considered problem, calling for methodological advances in longitudinal human behavior modeling. At the same time, research on integrating LLMs in health applications (e.g., generic medical question answering (Singhal et al., 2023) and mental health prediction (Xu et al., 2024)) has flourished over the past years. For this reason, latest works (Liu et al., 2023; Englhardt et al., 2024; Kim et al., 2024) explored the efficacy of LLMs in this area using standard in-context learning (e.g., zero-shot and few-shot prompting) and finetuning techniques, showing they are indeed the state of the art in longitudinal human behavior modeling. We hereby further discuss these works in the following paragraph.

**Longitudinal Human Behavior Modeling with LLMs.**  To our best knowledge, the earliest work in this area was conducted by Liu et al. (2023). By experimenting on multiple self-curated toy datasets[8], the authors showed that with only few-shot prompting, their pretrained LLM is viable for various health prediction tasks e.g., heart rate measurement, atrial fibrillation detection, and mood score prediction. Following up this work, Englhardt et al. (2024) instead considered different zero-shot prompting strategies with commercial LLMs (e.g., GPT-3.5/4 and PaLM 2) for depression and anxiety classification tasks on GLOBEM. Despite yielding low accuracy in clinical screening context, these general-purpose LLMs, even without any annotated examples provided, still generally outperformed the considered baseline ML models that required training from scratch on all labeled data. Lastly, Kim et al. (2024) expanded the experimentation from (Englhardt et al., 2024) by evaluating 12 cutting-edge LLMs with different in-context learning and finetuning techniques for 10 consumer health tasks derived from four ubiquitous health datasets (with GLOBEM and LifeSnaps included). They found that zero-shot prompting methods already yield comparable results to the considered supervised baselines (e.g., SVM, random forest, and classical pretrained language models such as BERT), and with proper few-shot prompting and finetuning techniques, the corresponding predictive performance can be improved significantly. All discussed works have thus demonstrated the compatibility of LLMs for longitudinal human behavior modeling. Nevertheless, they share the same nontrivial shortcomings as explained in our introduction, which severely undermine their implications for real-world health applications.

**Multi-Label Classification (MLC).**  Approaches to MLC are mainly categorized into algorithm adaptation and problem transformation (Gibaja & Ventura, 2015; Herrera et al., 2016). The latter approach, which transforms MLC problems to well-established single-label problems, has seen wider applications. Three most popular transformation methods are binary relevance (BR), classifier chain (CC), and label powerset (LP). BR and CC convert to multiple binary classification problems,

---

[8]The work was conducted prior to or during GLOBEM release and hence did not consider this dataset.

whereas LP converts to a multiclass classification problem. BR methods are often employed across various fields due to their simplicity and interpretability. However, unlike CC's, they omit the dependencies between labels altogether, which are in fact relevant in health and wellness applications (Wosiak et al., 2018; Blanco et al., 2020; Ge et al., 2020). On the other hand, CC methods are computationally expensive (Read et al., 2021) whereas LP's are prone to suffer from class-wise data scarcity (Tsoumakas & Vlahavas, 2007). Given the discussed underperformance of traditional ML methods and the promise of LLMs in longitudinal human behavior modeling, we approach MLC using ideas from the NLP literature (as elaborated in Sections 3.2 and 3.3), where the problem is better known as *multi-label text classification* (MLTC) (Chen & Ren, 2021).

Early notable works in MLTC that apply neural networks employ the sequence-to-sequence (Seq2Seq) model (Nam et al., 2017; Yang et al., 2018; 2019). Most recently, Lu et al. (2023) explore a hybrid model that combines CNN and LSTM for labeling short Chinese texts. Works that leverage LLMs, mostly via finetuning, are much rarer due to the extravagant resources required. Chang et al. (2020) attempt to finetune pretrained transformers such as BERT and XLNet for extreme MLTC (i.e., where the label collection is large), and Bețianu et al. (2024) propose a semi-supervised domain adaptation framework for BERT in order to ace MLTC tasks on target domains, to name a few. All aforementioned methods were benchmarked on generic text datasets comprising at least thousands of annotated samples. Therefore, we also seek to explore the uncharted feasibility of LLMs in MLTC problems on real-world ubiquitous health data.

## C  MORE ON EXPERIMENTS

### C.1  DATASETS

**LifeSnaps.**  Table S4 lists the features extracted from LifeSnaps. During data conversion, we divide the auxiliary data into three topics for the three respective questionnaires from the entry survey, namely TTM, BREQ, and IPIP. The negative/positive class ratios for tasks (i) and (ii) associated with this dataset are 99/31 and 86/44, respectively. Note that the definitions of "at risk" stated in Section 4.1 are for demonstration purposes only.

| Type | Feature |
|---|---|
| Time series $(D_{ts} = 8)$ | stress_score
lightly_active_minutes, moderately_active_minutes, very_active_minutes
sleep_efficiency
sleep_deep_ratio, sleep_light_ratio, sleep_rem_ratio |
| Auxiliary $(D_{ts} = 28)$ | (×11) Stages and Processes of Change Questionnaire (TTM)
(×6) Behavioural Regulations in Exercise Questionnaire (BREQ)
(×11) International Personality Item Pool version of the Big Five Markers (IPIP) |

Table S4: LifeSnaps features used.

**GLOBEM.**  Table S5 lists the features extracted from GLOBEM. During data conversion, we divide the 28-day time-series data into four topics (i.e., "first week", "second week", "third week", and "fourth week") and the auxiliary data into four topics as grouped in Table S5 (i.e., social well-being, mental well-being, physical health, and personality, respectively). Unlike LifeSnaps, the label definitions were respectively determined based on GLOBEM's percentile, the justifications from (Englhardt et al., 2024), and prior studies from multiple countries (Arnold et al., 2012; Warttig et al., 2013; Malik et al., 2020).

**CoSt.**  This dataset consists of three components. The first is an intake survey of 9 questions[9] asking for class standing, major, international student status, whether parent(s) went to college, highest education level of a parent/guardian and (if any) the other parent/guardian, total household income in the past 12 months, the extent of seeing themselves as a future engineer/scientist, and the perceived extent of others seeing them as a future engineer/scientist. The second component contains the students' grades of 17 lesser course items that were due within the first eight weeks: the first two diaries, six labs, four quizzes, and five homework assignments. Table S6 shows the chronology of these items. Collectively, these two components form the auxiliary data ($D_{aux} = 26$) and hence

---

[9]Excluded 2 questions regarding race and gender identity.

| Type | Feature |
|---|---|
| Time series ($D_{ts} = 16$) | date |
| | f_loc:phone_locations_doryab_totaldistance:allday |
| | f_loc:phone_locations_doryab_timeathome:allday |
| | f_loc:phone_locations_doryab_locationentropy:allday |
| | f_screen:phone_screen_rapids_sumdurationunlock:allday |
| | f_screen:phone_screen_rapids_avgdurationunlock:allday |
| | f_call:phone_calls_rapids_incoming_sumduration:allday |
| | f_call:phone_calls_rapids_outgoing_sumduration:allday |
| | f_blue:phone_bluetooth_doryab_uniquedevicesothers:allday |
| | f_steps:fitbit_steps_intraday_rapids_sumsteps:allday |
| | f_steps:fitbit_steps_intraday_rapids_countepisodesedentarybout:allday |
| | f_steps:fitbit_steps_intraday_rapids_sumdurationsedentarybout:allday |
| | f_steps:fitbit_steps_intraday_rapids_countepisodeactivebout:allday |
| | f_steps:fitbit_steps_intraday_rapids_sumdurationactivebout:allday |
| | f_slp:fitbit_sleep_intraday_rapids_sumdurationasleepunifiedmain:allday |
| | f_slp:fitbit_sleep_intraday_rapids_sumdurationawakeunifiedmain:allday |
| Auxiliary ($D_{ts} = 27$) | UCLA Loneliness Scale |
| | Sense of Social and Academic Fit Scale |
| | (×4) 2-Way Social Support Scale |
| | Everyday Discrimination Scale |
| | Chronic Work Discrimination and Harassment |
| | Perceived Stress Scale |
| | (×2) Emotion Regulation Questionnaire |
| | Brief Resilience Scale |
| | State-Trait Anxiety Inventory for Adults |
| | Center for Epidemiologic Studies Depression Scale—Cole version |
| | Beck Depression Inventory-II |
| | Mindful Attention Awareness Scale |
| | (×2) Brief Coping Orientation to Problems Experienced |
| | Gratitude Questionnaire |
| | Flourishing Scale & Psychological Well-Being Scale |
| | Cohen-Hoberman Inventory of Physical Symptoms |
| | The Brief Young Adult Alcohol Consequences Questionnaire |
| | (×5) The Big-Five Inventory-10 |

Table S5: GLOBEM features used.

correspond to two topics during our data conversion. The third component of CoSt comprises time-series data collected weekly via the ODIN app, with a total of 11 features ($D_{ts} = 11$) on students' self-evaluations and motivation (intrinsic and extrinsic) factors throughout the course as listed in Table S7. The negative/positive class ratios for tasks (i) and (ii) pertaining to this dataset are 38/10 and 36/12, respectively.

| Week | Item(s) |
|---|---|
| 1 | Lab 1 (due by Mon) |
| 2 | Lab 2 (Mon), Quiz 2 (Wed) |
| 3 | Homework 1 (Wed), Quiz 3 (Fri) |
| 4 | Lab 3 (Mon), Homework 2 (Wed), Lab 4 (Thu) |
| 5 | Diary 1 (Tue) |
| 6 | Quiz 4 (Mon), Lab 5 (Mon), Homework 3 (Wed), Lab 6 (Wed) |
| 7 | Homework 4 (Wed), Diary 2 (Thu) |
| 8 | Quiz 5 (Mon), Homework 5 (Wed) |

Table S6: Chronology of the course items' deadlines. There was a one-week Spring break between the end of Week 7 and the start of Week 8. Project 1 (not included) accounts for 10% of the final course grade and was due during the 10th week.

**PWUD.** The auxiliary data derived from the intake survey comprise individual attributes as well as substance use behaviors with respect to a wide variety of prescription and recreational drugs, including stimulants (methamphetamine, cocaine, amphetamines), depressants (benzodiazepines, opioids, heroin), and hallucinogens (marijuana, PCP, Ecstasy). Table S8 summarizes each feature group, each of which corresponds to a topic during our data conversion. Please refer to the attached ZIP file in our supplementary material for the complete list of survey questions. Table S9 lists the features collected daily via the ODIN app that constitute the 30-day time-series data. We create three topics accordingly i.e., "first 10-day", "second 10-day", and "third 10-day". The negative/positive

| Feature | Type | Question | Rule |
|---|---|---|---|
| Q1 | single-select | What grade do you think you might earn in CS1? | Saturday at 12:01 PM |
| Q2 | single-select | How much confidence do you have in your ability to complete all the requirements of the CS1 class? | Follow-up from Q1 |
| Q3 | single-select | How much confidence do you have in your ability to excel in the CS1 class? | Follow-up from Q2 |
| Q4 | single-select | How satisfied are you with your performance in this class? | Saturday at 07:00 PM |
| Q5 | single-select | How do you think other students in CS1 are performing compared to you? | Follow-up from Q4 |
| Q6 | single-select | How worried are you about our performance in the CS1 class? | Follow-up from Q5 |
| Q7 | single-select | How much do you see yourself as a future engineer or scientist? | Sunday at 12:01 PM |
| Q8 | single-select | How much do you think others (family, friends, peers) see you as a future engineer or scientist? | Follow-up from Q7 |
| Q9 | single-select | How important do you think the CS1 class is for your future career? | Follow-up from Q8 |
| Q10 | single-select | Thinking towards the future, how important do you think doing well in your college classes is to having a good life? | Follow-up from Q9 |
| Q11 | single-select | What type of on-campus extracurricular activities are you involved in? | Sunday at 07:00 PM |

Table S7: CoSt's time-series data. "multi-select" features are derived from check-all-that-apply questions.

class ratios for tasks (i)–(vi) (defined as suggested by domain experts in substance use research in our team) associated with this dataset are 23/32, 29/37, 18/49, 51/16, 50/18, 58/10, respectively. Note that the total counts vary across tasks due to missing data required for forming the corresponding labels in the exit survey. When computing loss and performance measures, if a sample has $Q' < Q$ missing labels, we simply ignore them and only consider the performance on the remaining $Q - Q'$ labels.

| Feature group/Topic | Summary of features |
|---|---|
| Social support (9/9) | -people the individual can confide in, obtain emotional support, and provide financial assistance
-perceived number of confidants and number of drug co-users |
| Substance use behaviors (19/21) | -frequency of smoking cigarettes, drinking alcohol, binge drinking, and using various non-injection drugs
-frequency of and reasons for using alcohol and/or drugs right before or while having sex |
| Drug overdose (13/15) | -experiences of overdoses (e.g., personal overdose history and awareness of Narcan/naloxone) |
| Substance use treatment (15/17) | -individual participation in alcohol and drug treatment programs (e.g., outpatient and detox)
-length of longest cessation period and count of relapse instances |
| Feelings (24/25) | -frequency of feeling nervous, restless, worried, and annoyed
-history of being diagnosed with various mental health disorders e.g., depression, ADHD, PTSD |
| Recent adverse experiences (10/10) | -whether being stolen, beaten up, robbed, threatened, physically and sexually assaulted in the past 30 days
-whether being incarcerated in the past 6 months and if so, whether it was drug related |
| Community views of substance use (8/8) | -community's perception regarding trustworthiness, harmfulness, accountability, and laziness of people who use cocaine/methamphetamine and opioids/heroin |
| Demographics (10/16) | -age, marital status, education, current employment status, religion, influence of religious or spiritual beliefs, ever been homeless, total household income
-whether experiencing job loss or pay cut due to COVID-19 |

Table S8: Feature groups in our intake/exit survey data. In parentheses next to each feature group, we provide [# of features we used from it]/[# of features it contains initially].

## C.2 IMPLEMENTATION DETAILS AND EXPERIMENTAL SETUPS

All experiments were conducted under Ubuntu 20.04 on a Linux virtual machine equipped with NVIDIA GeForce RTX 3050 Ti GPU and 12th Gen Intel(R) Core(TM) i7-12700H CPU @ 2.3GHz. We used PyTorch 1.13, CUDA 11.7, OpenAI 1.23, and scikit-learn 1.3. Code for MuHBoost is publicly available[10]. We employed GPT-3.5 (Brown et al., 2020) via OpenAI's Completions API (gpt-3.5-turbo-instruct) instead of GPT-3 (gpt-3-curie) as utilized in Summary-Boost, which as of 2024 is deprecated. We opted not to consider GPT-4 since it has been empirically shown in the work of SummaryBoost that larger GPT models do not consistently improve upon smaller models. While GPT-4 may exhibit stronger predictive performance than GPT-3.5 for zero-shot and few-shot settings (Kim et al., 2024), we show in Table 3 that the improvements are negligible when aggregating weak learners via boosting. Hence, in practice, the extra incurred monetary cost ($\times 6.7$[input]/$\times 15$[output] for GPT-4 and $\times 3.3$/$\times 7.5$ for the newly released GPT-4o with respect to GPT-3.5) from calling larger PLMs may outweigh the added benefits.

---

[10]https://github.com/AnonMouse3005/MuHBoost

| Feature | Type | Question | Rule |
|---|---|---|---|
| TimeUsed | multi-select | Yesterday, did you use any substances in any of these time intervals? | 10:00 AM |
| RelUseSelf | single-select | How much would you say you used substances yesterday? | † |
| WhichUse | multi-select | What substances did you use yesterday? (check all that apply) | † |
| UseWith | multi-select | Who did you use with yesterday? (check all that apply) | † |
| AlcoholUsed | multi-select | Yesterday, did you drink alcohol in any of these time intervals? | |
| DrinkWith | multi-select | Who did you drink with yesterday? (check all that apply) | † |
| NumDrinks | single-select | How many drinks did you have yesterday? | † |
| NumInteract | fill-number | Enter the number of people you interacted with yesterday? You must enter a whole number (e.g. 0=no one, 1=one person, 2=two people, etc.) | |
| FracKnow | single-select | Of all the people you interacted with yesterday, how many knew that you sometimes use substances? | |
| RelUseOther | single-select | Think of the people you saw yesterday who might be using substances themselves. How much do you think that they use? | |
| CoUse | single-select | Think of the people you saw yesterday who might be using substances. Did you and any of them use substances together yesterday? | |
| StressWhoUse | single-select | Think about any stressful interactions you had yesterday. Which of these statements is most true? | 05:00 PM |
| StressWhoKnow | single-select | Think again about the stressful interactions you had yesterday. Which of these statements is most true? | |
| StressExtent | single-select | Think about the stressful interactions you had yesterday. On a scale of 0 to 10, how stressful were they? | |
| Victim | multi-select | Did any of the following happen to you yesterday: (check all that apply) | |
| SupportWhoUse | single-select | Think about people who have helped you or who have been supportive today. Which of these statements is most true? | 08:00 PM |
| SupportWhoKnow | single-select | Think again about people who have helped you or who have been supportive today. Which of these statements is most true? | |
| SupportExtent | single-select | Think about the supportive interactions you had today. On a scale of 0 to 10, how supportive were they? | |
| AnySupport | single-select | Did you receive any support today? | |
| TypeSupport | multi-select | What type of support did you receive today? (check all that apply) | |
| TypeSupportOther | fill-text | Please list what other type of support you received. | † |
| Nervous | single-select | Today I felt nervous or anxious. (Agree or not?) | |
| Annoyed | single-select | Today I was easily annoyed or irritable. (Agree or not?) | |

Table S9: Features collected daily via ODIN app in chronological order. "multi-select" features are derived from check-all-that-apply questions. (†) denotes questions that can be skipped from skip logic: If (i) **TimeUsed** is "I did not use any substances yesterday", (ii) **AlcoholUsed** is "I did not drink alcohol yesterday", and (iii) **TypeSupport** is *not* "Other".

For data conversion, we limit each data description to 200 words in order to accommodate as many data descriptions in the summarization prompt as possible[11]. As opposed to SummaryBoost, we do not preprocess numerical data (where they are supposedly binned into percentiles and encoded descriptively e.g., "low", "medium", "high") to ensure minimal information loss, especially for numerical time series. We believe this omission can be counterbalanced by the fact that SummaryBoost originally employed GPT-3's `Curie`, a comparatively small and now deprecated model, and that bigger LLMs have shown much more encouraging capability in numerical reasoning (Englhardt et al., 2024). The number of boosting rounds $T$ is set to a large value of 100 for training till convergence (typically within 10–20 rounds) and the size of the representative subset $s$ is set as large as possible[11] (without exceeding the maximum context length) to 10. We set $\mu$, the nonzero hyperparameter for raising the bar for each weak learner, to 0 since we notice no significant increases in predictive performance otherwise. The stopping threshold at each round for both MuHBoost and MuHBoost[LP+], defined by $1 - 1/K - \mu$, is hence $1 - 1/\min\{N, 2^Q\}$ (below which is considered satisfactory and no further resampling is needed in accordance with SummaryBoost). For MuHBoost[CC] ($K = 2$), we introduce the discount factor $\gamma = 0.95$ into this threshold, which becomes $1 - (1/2)\gamma^q$ for each label $q \in [0, Q-1]$, to relax the training further down the chain (i.e., subject to error propagation). For the permutation function $\sigma$ required in this MuHBoost variant, we order the $Q$ labels by descending frequency (i.e., labels with high proportion of positive samples come first), which has been shown to yield decent predictive performance (to be discussed shortly) among other heuristics (Wang et al., 2016; Nam et al., 2017).

**Baselines.** We consider the following methods from both problem transformation (•) and algorithm adaptation (∗) approaches for MLC.

---

[11]These configurations were shown to universally improve predictive performance according to the ablation studies from SummaryBoost.

- *Zero-shot*: Similarly as in the work of SummaryBoost, we query the LLM with a data description and directly infer its label. For MLC, this can be done in three ways: asking for either (i) $Q$ labels one at a time (hence $Q$ separate calls to the LLM), (ii) $Q$ labels simultaneously, or (iii) only its positive labels, which are reminiscent of BR, LP, and LP+ approaches, respectively. We therefore denote them as **0-shot[BR]**, **0-shot[LP]**, and **0-shot[LP+]**. Note that for single-label problems ($Q = 1$), 0-shot[BR] is equivalent to the zero-shot prompting techniques developed in (Englhardt et al., 2024; Kim et al., 2024). We use GPT-4 (`gpt-4-turbo`) given its superior performance compared to GPT-3.5 as shown by Kim et al. (2024).

- *Few-shot*: Given the good performance of few-shot prompting methods considered in (Liu et al., 2023; Kim et al., 2024), we also include this baseline in our evaluations. We follow the few-shot setting as described in SummaryBoost's experiments, where we leverage their `ClusterSampling` to select a representative subset of $\mathbf{X}_{train}$ for demonstration before asking the LLM to infer the label of some unseen sample. We set the number of shots to 10, which is consistent with $s$ fixed earlier. Similar to the zero-shot setting, there are three ways to ask the LLM for MLC, for which we denote as **10-shot[BR]**, **10-shot[LP]**, and **10-shot[LP+]**. We also use GPT-4 for the same stated reason.[12]

Additionally, we consider the following methods from classical ML algorithms. Given the various challenges in working with ubiquitous health data as discussed in Section 1 and the current lack of dedicated representation learning techniques for them to our best knowledge, following the benchmarking of SummaryBoost, we instead embed the data descriptions via OpenAI's embeddings API (`text-embedding-ada-002`), which are then fed into these baselines as input features.

- *Random forest* (Ho, 1995) and *XGBoost* (Chen & Guestrin, 2016): We consider problem transformation via both CC and LP when employing these baselines for MLC, for which we denote respectively as **RF[CC]**, **RF[LP]**, **XGBoost[CC]**, and **XGBoost[LP]**.

* *Multi-label $k$ Nearest Neighbours* (**ML$k$NN**) (Zhang & Zhou, 2007) and *twin multi-label SVM* (**MLTSVM**) (Chen et al., 2016): Lastly, we include two well-established methods for MLC via algorithm adaptation, which respectively adapts $k$NN and SVM to MLC directly without requiring problem transformations.

**Evaluation Metrics and Procedure.** For each considered set of experimental configurations, we used the split ratio of 50/10/40 (for train/validation/test set), which follows SummaryBoost's evaluation, for a total of 10 different splits. It has been shown that conventional stratified sampling approaches are not effective when partitioning multi-label datasets (Sechidis et al., 2011) into training (train+validation) and test sets. We therefore adopt the *iterative stratification* algorithm[13] proposed in the same cited work, which solves this issue by considering each label separately, starting from the one with fewest positive examples (i.e., most *rare* label) and working its way to the best represented one (i.e., most *frequent* label).

Since we work with MLC problems having small to moderate $Q$, we can use the standard *Hamming accuracy* (HA) $= \frac{1}{Q} \sum \mathbb{1}\{\hat{\boldsymbol{y}} = \boldsymbol{y}\} \in [0, 1]$ metric for MLC (Tsoumakas & Katakis, 2007), where $\boldsymbol{y}, \hat{\boldsymbol{y}}$ are vectors of ground-truth labels and their corresponding predictions. It can be described as the fraction of correctly predicted labels to $Q$ given a sample, and hence we report the resulting summed-then-averaged value from all test samples. Since we encounter class imbalance for individual labels in 3 out of 4 datasets, we also consider so-called *label-based* metrics, namely *micro-* and *macro-averaged-$F_1$-measure* (Herrera et al., 2016). They are respectively defined by Nam et al. (2017) as $\text{mi}F_1 = \frac{\sum_{q=1}^{Q} 2\text{TP}_q}{\sum_{q=1}^{Q}(2\text{TP}_q + \text{FP}_q + \text{FN}_q)}$ and $\text{ma}F_1 = \frac{1}{Q} \sum_{q=1}^{Q} \frac{2\text{TP}_q}{2\text{TP}_q + \text{FP}_q + \text{FN}_q}$, where TP, FP, FN stand for true positives, false positives, and false negatives from all test samples. Whereas the former evaluates the overall predictive performance of a classifier (and hence rewards those giving correct predictions on frequent labels), the latter focuses on assessing performance on rare labels.

---

[12]Note that we did not include the finetuning technique from (Kim et al., 2024) in our baselines, which reportedly yielded the overall best performance, due to the fact that the employed finetuned model has yet been released at the time of writing this paper.

[13]`https://github.com/scikit-multilearn-ng/scikit-multilearn-ng` (Szymański & Kajdanowicz, 2017).

## C.3 RESULTS

**MuHBoost vs. MuHBoost[LP+].** Even though MuHBoost and MuHBoost[LP+] only differ in the inference directive in essence, such modification has nontrivial impact as highlighted in Table S10 (which extends Table 1). In particular, MuHBoost[LP+] generally outperforms its predecessor, and the improvement in predictive performance is most noticeable when the number of labels $Q$ is large (for PWUD dataset). This observation supports our claim in Section 3.3 that LLMs may hallucinate when being asked to answer multiple related questions simultaneously. In response, MuHBoost[LP+] alleviates this issue by simplifying the inference prompt, where instead of asking for an array of 'Yes' and 'No' in the exact order (e.g., listed in [DRUG LIST] from Figure 2), it only requests the names of *positive* labels in any order. (MuHBoost[CC] achieves the same goal of alleviating hallucinations during inference, albeit by reformulating the MLC problem into multiple binary classification problems instead.) This greatly reduces the chance of inconsistencies between the output prediction array from LLMs and the corresponding input list of labels (e.g., [DRUG LIST]).

| Method | LifeSnaps+ | GLOBEM+ | CoSt+ | PWUD+ |
|---|---|---|---|---|
| MuHBoost | **0.734(44)** | 0.712(51) | 0.701(49) | 0.625(61) |
| | **0.837(53)** | 0.824(60) | 0.796(56) | 0.734(63) |
| | 0.818(47) | 0.815(63) | 0.785(54) | 0.710(59) |
| MuHBoost[LP+] | 0.731(29) | **0.720(43)** | **0.715(38)** | **0.688(54)** |
| | 0.836(46) | **0.829(62)** | **0.814(54)** | **0.784(48)** |
| | **0.822(40)** | **0.817(59)** | **0.803(49)** | **0.771(41)** |

Table S10: Detailed predictive performance of MuHBoost vs. MuHBoost[LP+] from Table 1. In each cell: (top) HA, (middle) mi$F_1$, (bottom) ma$F_1$.

| LifeSnaps+: Perform data conversion? | | |
|---|---|---|
| No | Yes | |
| | Refine data description? | |
| | No | Yes |
| 0.625(58) | 0.709(45) | **0.731(29)** |
| 0.667(64) | 0.806(62) | **0.836(46)** |
| 0.649(57) | 0.784(57) | **0.822(40)** |

Table S11: Ablation of the (entire) data conversion procedure on LifeSnaps+ using MuHBoost[LP+] with GPT-4.

**Further Ablation.** We ablate the data conversion procedure (in addition to the abstractive summarization step previously conducted in Table 2) to better understand its overall impact on predictive performance. Because the set of representative samples from raw (serialized) data[14] in the summarization prompt can easily exceed the context length of GPT-3.5, we use GPT-4 (with $30\times$ longer context length). More specifically, we perform this ablation using MuHBoost[LP+] on LifeSnaps (with auxiliary data included). As shown in Table S11, there is a sharp decline in predictive performance when prompting the LLM using raw data. This result demonstrates the importance of our data conversion procedure, which faithfully extracts the most relevant information from a record and subsequently mitigates hallucinations in the LLM during model training and inference.

**Additional Benchmarking.** We include another baseline, AdaRNN[15] (Du et al., 2021), which was recommended by the authors of GLOBEM as a potential solution to longitudinal human behavior modeling. As shown in Table S12, MuHBoost (and hence its variants) still outperforms this method by a considerable margin, which again highlights the efficacy of our proposed methods.

| Method | LifeSnaps+ | GLOBEM+ | CoSt+ | PWUD+ |
|---|---|---|---|---|
| AdaRNN | 0.656(41) | 0.628(39) | 0.614(33) | 0.543(40) |
| | 0.742(46) | 0.735(44) | 0.708(37) | 0.636(52) |
| | 0.730(51) | 0.729(41) | 0.691(46) | 0.608(49) |
| MuHBoost | **0.734(44)** | **0.712(51)** | **0.701(49)** | **0.625(61)** |
| | **0.837(53)** | **0.824(60)** | **0.796(56)** | **0.734(63)** |
| | **0.818(47)** | **0.815(63)** | **0.785(54)** | **0.710(59)** |

Table S12: Predictive performance of MuHBoost vs. AdaRNN. In each cell: (top) HA, (middle) mi$F_1$, (bottom) ma$F_1$.

**Computing, Time, and Monetary Resource Consumption.** MuHBoost requires minimal computing power by leveraging state-of-the-art LLMs via commercial APIs similarly as SummaryBoost

---

[14]Recall that the extractive summarization step in our data conversion procedure effectively trims each record within a reasonable word limit.

[15]https://github.com/jindongwang/transferlearning/tree/master/code/deep/adarnn.

does, which has the advantages of being gradient-free and operating without accessing and tuning the LLM's internal parameters (Manikandan et al., 2023). This property is particularly useful for domain experts to quickly incorporate the method into their usual practices. Since calling these LLMs costs a certain amount as well as taking arbitrary runtime[16], it is important to analyze the time complexity and expected cost of MuHBoost and its variants[17]. First, MuHBoost and [LP+] share the complexity with SummaryBoost and therefore take $O(TMf)$, where $T$ is the number of boosting rounds, $M$ is the number of resampling for the weak learner at each round, and $f$ is the LLM's runtime (see Appendix D.5 for an analysis). In terms of cost associated with training a model via MuHBoost or [LP+], according to the estimation by Manikandan et al. (2023), the upper bound for the total number of requested tokens is $n_t = T \times [M \times (S_t + 0.5N \times P_t) + 0.1N \times P_t]$, where $S_t$ and $P_t$ are the upper bounds of input tokens for summarization and inference ($S_t \approx P_t \times s$), respectively, and 0.5 and 0.1 are the split percentages chosen for the train and validation sets. For [CC], given that the CC approach transforms an MLC problem with $Q$ labels into $Q$ single-label classification problems, a naïve complexity analysis would simply add an extra factor of $Q$ for both time and cost. However, recall that the adoption of AdaBoost.C2 as backbone for [CC] (Algorithm 2) allows asynchronous training for each label, which means it can be easily stopped in advance and excluded from further boosting rounds when needed. This neat property hence effectively reduces the extra factor to $Q' \leq Q$. For a comparison with the state of the art in longitudinal human behavior modeling, the finetuning approach from (Kim et al., 2024) (exhibiting the overall best performance) takes $O(TNf)$ for *one* task, where $T$ is the number of epochs, or $O(QTNf)$ in total for an MLC problem. Thus, MuHBoost offers a resource-efficient as well as effective solution for modeling ubiquitous health data.

**[LP+] or [CC]?** Given the overall best predictive performance of both MuHBoost variants, we summarize cases when one may be more favorable than the other in Figure S3 below.

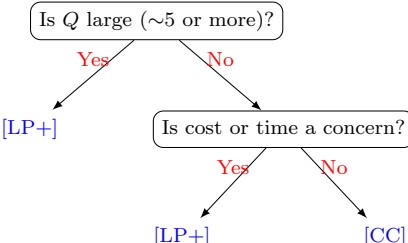

Figure S3: Decision tree for considering the two MuHBoost variants.

**Interpretability.** Following SummaryBoost, MuHBoost and its variants also provide interpretability by storing LLM-generated knowledge regarding the task at hand into natural-language summaries. By this means, domain experts can directly inspect the factors that drive the LLM's predictions by looking at these summaries and their associated coefficients in all boosting rounds, which is valuable in practice for filtering erroneous summaries (done simply by tuning their coefficients to 0).

# D MORE ON MUHBOOST

## D.1 ILLUSTRATIVE EXAMPLE OF DATA CONVERSION PROCEDURE

Figure S4 provide an illustrative example of our data conversion procedure for LifeSnaps, which follows the template introduced in Figure 1. Note that since the time-series data from LifeSnaps contains only seven time points, we do not divide the record into multiple topics/subperiods.

---

[16]There could be intermittent delays when calling GPT due to the API's rate limits (https://platform.openai.com/docs/guides/rate-limits).

[17]Vanilla MuHBoost and MuHBoost[LP+] follow the same analysis.

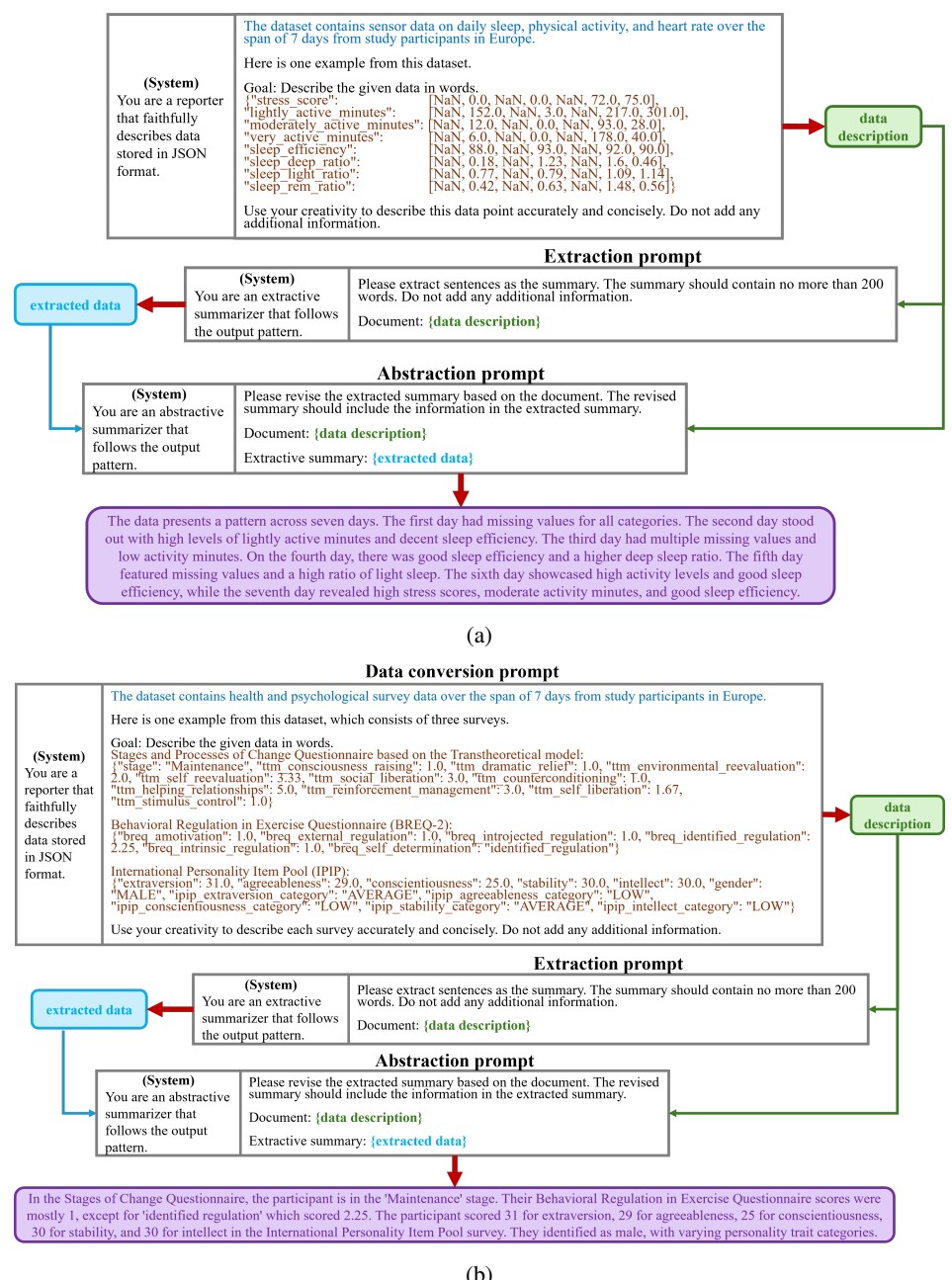

Figure S4: Conversion of (a) time-series and (b) auxiliary data from LifeSnaps. The purple box contains the respective refined data description.

## D.2 MUHBOOST[CC]

**Motivations and Design Objectives.** Conceptually, CC approaches MLC problems by linking $Q$ binary classifiers in a 'chain', such that the output prediction of one classifier is appended as an additional feature to the input of all subsequent classifiers (Read et al., 2011). By this means, unlike BR, CC can model relationships between labels and hence yield improved predictive performance over BR approaches (Read et al., 2021). Compared to LP, CC is more reliable in practice since it does not suffer as much from the aforementioned rare label issue and can generalize to the case of *multiclass* MLC (i.e., some or all of the $Q$ labels are multiclass) (Kocev et al., 2007; Read et al., 2014), in exchange for being more computationally expensive (by a factor of $Q$). CC is also prone to

the error propagation issue (i.e., the errors produced in the front may propagate down the chain and bring further errors, especially as the chain gets longer) and raises additional questions on how to best order the labels in the chain (Read et al., 2021). Therefore, we aim to address these bottlenecks when developing MuHBoost[CC].

**Chain Order.** For determining the label ordering, AdaBoost.C2 heuristically reorders the labels in the chain at each boosting round by ascending training error rate. This aims to alleviate the error propagation issue in exchange for additional bookkeeping of the order matrix $\mathbf{O}$, which is later accessed during inference. Formally, let $H_t = \mathbf{H}[t,:]$ be the classifier chain at round $t$ whose order is given by $O_t = \mathbf{O}[t,:]$. During training, the prediction of $H_t$ given a training sample $\boldsymbol{x}$ is an array of length $Q$, wherein $\hat{y}^{t,q}$ is the atomic prediction for its $q$th binary label $y^q$ at round $t$ and can be expressed as

$$\hat{y}^{t,q} = \mathbf{H}[t,q](\boldsymbol{x}^{o-1}),$$
$$\boldsymbol{x}^{o-1} = \boldsymbol{x}^{o-2} \oplus y^{o-1}, \tag{S1}$$
$$\boldsymbol{x}^1 = \boldsymbol{x}.$$

Here, $o = \mathbf{O}[t,q]$ is the index of $q$ in $O_t$, $o-1 = \mathbf{O}[t,q-1]$ is the index of the label $q'$ that immediately precedes $q$ in $O_t$ (i.e., $\epsilon[q'] < \epsilon[q]$), and so on. At inference time, the final prediction for $y^q$ given an unseen sample is then produced by a weighted majority vote from the predictions of the classifiers $\mathbf{H}[:,q]$ in all boosting rounds, where the corresponding weights are $\mathbf{A}[:,q]$. On the one hand, this method severely reduces interpretability since each round $t$ may potentially yields a different ordering $O_t$, requiring extra efforts to interpret all $Q$ predictions especially for larger boosting rounds $T$. On the other hand, its effectiveness is dubious as no related ablation studies were conducted (Li et al., 2023). Furthermore, Read et al. (2021) challenged this idea by demonstrating how poor predictions for a particular label may potentially serve as good feature expansions for other labels' predictions. Combined with the fact that chain ordering is still an open problem (Scanagatta et al., 2019), following (Nam et al., 2017), we therefore simply opt for a reasonable chain order defined by some heuristic permutation $\sigma$ and fix it throughout the boosting process. The first two lines of Equation S1 can hence be simplified as

$$\hat{y}^{t,q} = \mathbf{H}[t,q](\boldsymbol{x}^{\sigma(q-1)}),$$
$$\boldsymbol{x}^{\sigma(q-1)} = \boldsymbol{x}^{\sigma(q-2)} \oplus y^{\sigma(q-1)}. \tag{S2}$$

Note that in addition to chain ordering, we also order the labels according to $\sigma$ for the label information concatenated to each data description as well as the e.g., [DRUG LIST] in the summary and inference directives (Figure 2). The same applies to MuHBoost and MuHBoost[LP+].

### D.3 ONLINE BOOSTING

To ensure MuHBoost[18] can efficiently update from the influx of new ubiquitous health data, we demonstrate how it can be extended to *online learning* settings as shown in Algorithm 3. Let $R$ and $\beta_r$ denote the number of boosting rounds and the coefficient of a weak learner at round $r \leq R$, respectively, in this online learning setting. The subroutine `Collate` incorporates new data into the existing dataset, which can be from either newcomers or additional time points for the same $N$ participants. Note that our previously discussed data conversion step can accommodate both types of new data. After re-initializing the weights of the samples (Step 5), we select their representative subset via `ClusterSampling` and use it to build $T$ up-to-date weak learners similarly as in SummaryBoost in order to enrich the existing pool of $T$ weak learners, $H$, which is obtained from the last checkpoint of MuHBoost (Steps 6–9). Then, Steps 10–15 constitute the online boosting procedure inspired by (Grabner et al., 2006), where at each round $r$ we first select the best weak learner $h_t^*$ in the pool subject to its training error rate (which depends on the sample weights $\mathbf{w}$), followed by computing $\beta_r$, then updating $\mathbf{w}$. The returned set of $R$ weak learners, $\mathcal{H}$, would serve as the pool $H$ in the next update. Notice this simple extension takes only $O(Rf)$ instead of $O(RMf)$ if we were to retrain MuHBoost from scratch[19], where $M$ is the number of resampling (of summaries) per round and $f$ is the LLM's runtime in accordance with (Manikandan et al., 2023).

---

[18]Applies to its variants as well.
[19]MuHBoost[CC] requires another factor of $Q$ for both.

---

**Algorithm 3 Online MuHBoost**

---

**Input:** $\mathbf{X}_{train}$, new data $\mathbf{X}_{new}$, pool of weak learners $H = \{h_1, \ldots, h_T\}$ returned from last checkpoint of MuHBoost (i.e., trained on $\mathbf{X}_{train}$), number of boosting rounds $R$.
1: $\mathcal{H}, \mathcal{B} \leftarrow$ new empty arrays each of length $R$
2: $H' \leftarrow$ new empty array of length $T$
3: $\mathbf{X}_{train} \leftarrow \texttt{Collate}(\mathbf{X}_{train}, \mathbf{X}_{new})$
4: $N \leftarrow |\mathbf{X}_{train}|$
5: $\mathbf{w} \leftarrow$ new array of length $N$ filled with $\frac{1}{N}$
6: **for** $t = 1$ to $T$ **do**
7:     $\mathbf{X}_s \leftarrow \texttt{ClusterSampling}(\mathbf{X}_{train}, \mathbf{w}, \ldots)$
8:     $H'[t] \leftarrow \texttt{Summary}(\mathbf{X}_s)$
9: $H \leftarrow H \cup H'$                                                       ▶ a total of $T \times 2$ weak learners in the pool
10: **for** $r = 1$ to $R$ **do**
11:     Estimate error rate $\epsilon_t$ for each $h_t \in H$ on $\mathbf{X}_{train}$ using $\mathbf{w}$
12:     $h_t^* \leftarrow h_t$ that yields lowest $\epsilon_t$
13:     Compute coefficient $\beta_r$ for $h_t^*$ using $\epsilon_t$
14:     Update $\mathbf{w}$ using $h_t^*$ and $\beta_r$
15:     $\mathcal{H}[r] \leftarrow h_t^*$;   $\mathcal{B}[r] \leftarrow \beta_r$
**Return:** $\mathcal{H}, \mathcal{B}$

---

This step can be repeated multiple times as needed. For instance, if there are additional topics such as newly collected temporal data, we can simply prompt the LLM again to describe the e.g., "fourth 10-day" data.

### D.4 SUMMARY OF HOW MUHBOOST IS COMPATIBLE WITH UBIQUITOUS HEALTH DATA

Recall that ubiquitous health data brings three additional challenges to existing ML approaches for TSC. Our proposed methods, MuHBoost and its variants, address them and stand out from existing LLM-based approaches (Liu et al., 2023; Englhardt et al., 2024; Kim et al., 2024) as follows:

1. **Long time series.** MuHBoost divides long time series into multiple smaller chunks called 'topic' using an automated procedure that leverages zero-shot summarization capabilities of advanced LLMs, whereas the aforementioned LLM-based approaches simply fit the entire record into the prompt, which might not be feasible given the employed LLM's context length or would result in a decline in performance (Gruver et al., 2024; Liu et al., 2024).

2. **High rate of missing values.** Although all LLMs can naturally accommodate any amount of missing values (by denoting missingness with text placeholders such as '-' or 'NaN'), for time-series data in particular, they may encounter difficulties interpreting the temporal pattern of multiple features for a record when its missing pattern is arbitrary (Figure S5b). Given that all cited LLM-based approaches fit the entire record into the prompt with minimal data preprocessing, such shortcoming from LLMs applies to these approaches as well. In contrast, MuHBoost prompts the employed LLM to describe each record in natural language, which directly focuses on the temporal pattern of each feature and hence can effectively deal with any data missing pattern.

3. **(Very) Small sample size.** LLMs can effectively deal with time-series data of small sample size from their encoded prior knowledge (Gruver et al., 2024), and hence all LLM-based approaches including ours have shown to work well on small datasets. However, existing approaches were not evaluated on more general ubiquitous health datasets with *heterogeneous* time series commonly encountered in longitudinal human behavior modeling.

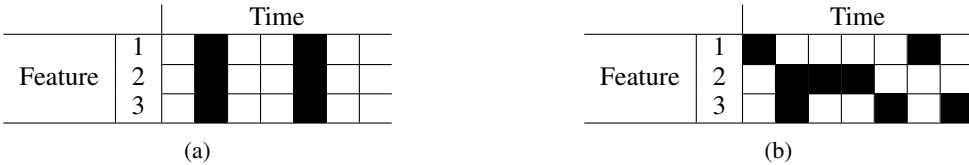

Figure S5: Example of (a) non-arbitrary vs. (b) arbitrary data missing pattern in multivariate time-series data, where each black cell denotes a missing value.

### D.5 COMPLEXITY ANALYSIS AND FLOPs ESTIMATION

**Time Complexity.** Algorithm 4 shows the complete procedure for training MuHBoost and MuH-Boost[LP+], which follows the boosting process of SummaryBoost (Algorithm 2 in (Manikandan et al., 2023)). At boosting round $t \in \{1, \ldots, T\}$, a candidate summary or weak learner $\mathbf{h}[t]$ for the representative set of data descriptions $(\mathbf{X}_s, \mathbf{y}_s)$ is sampled by prompting the LLM (Step 5). Then, we compute the training error $\epsilon$ for $\mathbf{h}[t]$ to determine if it performs at least better than random guess-ing (Step 6), which involves calling the LLM $N$ times for inference. Let $f$ be the arbitrary runtime[16] of the LLM for both summarization and inference. If $\epsilon$ is not satisfactory, we repeat the process (Steps 7–8) and hence resample $\mathbf{h}[t]$ for a total of $M \geq 1$ times. Once the desired weak learner is found after $O(Mf)$, we compute its coefficient (Step 9), recompute and re-normalize the sample distribution $\mathbf{w}$ (Steps 10–11), and compute the validation error for round $t$ only once. Thus, the final complexity is $O(TMf)$.

---

**Algorithm 4 MuHBoost and MuHBoost[LP+]**

**Input:** Training data $\mathbf{X}_{train}$ with sample size $N$, label array $\mathbf{y}_{train}$ of length $N$, maximum number of boosting rounds $T$, size of the representative subset $s$, hyperparameter $\mu \in [0, 0.5)$.

1: $\mathbf{h}, \mathbf{a} \leftarrow$ new empty arrays of length $T$
2: $\mathbf{w} \leftarrow$ new array of length $N$ filled with $\frac{1}{N}$
3: **for** $t = 1$ to $T$ **do**
4:     $(\mathbf{X}_s, \mathbf{y}_s) \leftarrow$ ClusterSampling$(\mathbf{X}_{train}, \mathbf{y}_{train}, \mathbf{w}, s)$    ▶ Algorithm 1
5:     $\mathbf{h}[t] \leftarrow$ Summary$(\mathbf{X}_s, \mathbf{y}_s)$    ▶ Summary prompts the LLM to summarize the representative samples
6:     $\epsilon \leftarrow \frac{\sum_{i=1}^{N} \mathbf{w}[i] \times \mathbb{1}\{\mathbf{h}[t](\mathbf{X}_{train}[i]) \neq \mathbf{y}[i]\}}{\sum_{i=1}^{N} \mathbf{w}[i]}$    ▶ $\epsilon$ is the training error rate
7:     **if** $\epsilon \geq 0.5 - \mu$ **then**    ▶ $\mu \geq 0$ enforces higher-quality summaries for accelerating convergence
8:         Go to Step 4
9:     $\mathbf{a}[t] \leftarrow \log\left(\frac{1-\epsilon}{\epsilon}\right)$
10:     **for** $i = 1$ to $N$ **do**
11:         $\mathbf{w}[i] \leftarrow \mathbf{w}[i] \times e^{\mathbf{a}[t] \times \mathbb{1}\{\mathbf{h}[t](\mathbf{X}_{train}[i]) \neq \mathbf{y}[i]\}}$
12:     $\mathbf{w} \leftarrow$ Normalize$(\mathbf{w})$

**Return:** $\mathbf{h}, \mathbf{a}$

---

**FLOPs Estimation.** MuHBoost queries $n_t$ tokens in total during training as previously discussed. Therefore, according to Kaplan et al. (2020) from OpenAI, it requires approximately $2n_{param} \times n_t$ FLOPs (i.e., add-multiply operations) to train MuHBoost[17], where $n_{param}$ is the number of LLM parameters (175 billions for GPT-3.5). Assuming we are dealing with a dataset consisting of $N = 100$ samples and we respectively allocate 50% and 10% of them for training and validation. In this case, $n_t$ would be 15 rounds $\times$ [10 resampling $\times$ (3072 summary tokens + (0.5 $\times$ 100 training samples) $\times$ 333 prediction tokens) + (0.1 $\times$ 100 validation samples) $\times$ 333 prediction tokens] $\approx$ 3 million tokens. Thus, training MuHBoost on this dataset takes $1.05 \times 10^{18}$ FLOPs or $1.05 \times 10^9$ GFLOPs, and training MuHBoost[CC] takes an additional factor of $Q'$.

### D.6 LIMITATIONS AND FUTURE WORK

MuHBoost and its variants were developed to deal with ubiquitous health datasets that typically have small sample size. Therefore, we do not expect our proposed methods to scale to larger datasets (i.e., with thousands of samples) in terms of resource efficiency. When the opportunity to work with large ubiquitous health data arises, approaches that require ample sample size such as LLM finetuning become more suitable than ours i.e., boosting LLM-generated weak learners. Regardless, some core ideas of our work can still apply to these approaches. More specifically, our data conversion procedure and prompt engineering techniques when developing MuHBoost and MuHBoost[LP+] for MLC can be leveraged to effectively (i) deal with heterogeneous time series and (ii) reduce resource consumption as discussed throughout the paper.

Although our data conversion procedure can deal with long time series, we only experimented on datasets with at most 30 time points (PWUD). We were not able to extend our experiments to longer time series due to the limitations of currently available ubiquitous health datasets. We believe the release of datasets that capture human behaviors at finer scales would further validate the efficacy of our methods in addressing this challenge.

### D.7 BACKGROUND ON SUMMARYBOOST

SummaryBoost was proposed as a novel approach to address tabular data classification under small sample size and high heterogeneity (i.e., containing different feature types, e.g., categorical and continuous as well as missing values) (Manikandan et al., 2023). Its novelty is two-fold. First, because it has been shown that data preprocessing is particularly important for deep learning approaches when working with tabular data Borisov et al. (2022), the majority of works in this area focus on extensive data preprocessing techniques. SummaryBoost, on the other hand, leverages the natural capabilities of LLMs in text summarization to convert tabular data records into natural language descriptions, which circumvents the difficulties stemmed from data heterogeneity altogether. Second, SummaryBoost incorporates LLMs into a traditional boosting framework (i.e., AdaBoost) by iteratively asking them to summarize a given set of representative samples from the (transformed) data, the output of which serves as a weak learner for the current boosting round. This approach is especially useful for small sample size settings (i.e., $\leq 100$ samples) since the prior knowledge encoded into LLMs (which has improved tremendously over the years) can be fully exploited, resulting in improved predictive performance as sufficient LLM-generated weak learners are included.

Because real-world ubiquitous health data share the same stated challenges as tabular data, we believe these two novel ideas from SummaryBoost can be applied to our problem.

