# OpenReview forum: "MuHBoost: Multi-Label Boosting For Practical Longitudinal Human Behavior Modeling"
_ICLR.cc/2025/Conference — ICLR 2025 Poster_

### Official Review · Reviewer_qBMd · 2024-10-26

**Soundness:** 4
**Presentation:** 2
**Contribution:** 3
**Rating:** 8
**Confidence:** 3

**Summary:**

The authors consider the task of using longitudinal ubiquitous health data to predict human behavior. They create a model, called MuHBoost, based on both LLMs (SummaryBoost model methodology) and multilabel classification (label powerset and classifier chain methods). The authors compare their model to various baselines and show that MuHBoost outperforms all baselines and presents two key advantages: (i) MuHBoost can consider different feature types and missing values and (ii) MuHBoost requires less computing time and cost. To address LLM hallucination the authors also build two extensions of their base model (which result in better performance). The authors apply their model to 4 datasets to predict psychology, student performance, and drug use.

**Strengths:**

S1 - Important application. The authors investigate an important application of longitudinal human behavior modeling with applications to patient monitoring, interventions for drug users, etc.

S2 - Multiple datasets: The authors evaluate their model on multiple, diverse datasets with a range on the number of labels (ranging from 2 to 6 labels).

S3 - Multiple baselines and metrics: The authors compare their model to a diverse set of baselines and evaluate on multiple standard MLC metrics.

S4 - Multiple robustness checks: The authors run various robustness checks including on SummaryBoost’s extract-then-refine procedure and on the use of GPT4 (more resource expensive) vs GPT3.5 (less resource expensive).

**Weaknesses:**

W1 - Needs more background clarification: The main weakness of the paper is a lack of background clarification. The authors should give more clarification on the SummaryBoost, LP and CC methods which the MuHBoost model is built off of.

W2 - Comparison to multilabel text classification: The authors state that the use of LLMs for MLC type prediction tasks have been done in multilabel text classification works. How do these methods compare to MuHBost?

W3 - Statistical significance: Are the results in table 1 statistically significant?

W4 - Only binary class predictions: The authors claim that MuHBoost presents an advantage over prior work because their model can consider different feature types. However, all experiments run in the paper only consider binary class predictions. Have the authors evaluated the model on other feature types as well (e.g. categorical/continuous prediction)?

**Questions:**

1. How do the authors combine data descriptions across topics?
2. For the two psychology datasets (LifeSnaps and GLOBEM), why are some tasks (e.g., negative emotions or anxiety) defined differently across datasets?
3. When describing baselines in Section 4.2, what do problem transformation and algorithm adaptation mean?

Minor comments / suggestions:

- Throughout I recommend using parenthetical citations (citep instead of cite)
- At multiple points, footnotes referenced in the main text were placed in the appendix. Could each footnote be placed on the page they are first referenced on?
- Line 081: “have the two following”
- Line 206: “require” instead of “requires”

---

> ### Author Response · Authors · 2024-11-23
>
> We thank Reviewer qBMd for your thorough review of our paper. We would like to respectfully present our rebuttal as follows.
>
> ## Weaknesses
>
> > W1 - Needs more background clarification
>
> We appreciate your concern regarding the background of SummaryBoost. To clarify, it was proposed to specifically address tabular data classification only with small sample sizes and high heterogeneity. Therefore, we do not include it in the background or related work, though we did clarify its relevance to ubiquitous health data in the Introduction (in Contribution I). For MLC approaches such as LP and CC methods, we introduced some background and discussed relevant works in Section 2 and Appendix B (under MLC subsection for both). We also clarified important works upon which our methods are built throughout Section 3 as well as Appendix D.1 (for MuHBoost[CC]).
>
> > W2 - Comparison to multilabel text classification
>
> Existing methods in multilabel text classification (MLTC) mentioned in the Related Work (last paragraph in Appendix B) have only been applied to generic text labeling tasks on large datasets (with thousands of samples). Furthermore, to our best knowledge, none employ commercial LLMs such as GPT (mostly on pretrained transformers such as BERT instead).
>
> > W3 - Statistical significance
>
> We ranked the considered methods in Table 1 by the average of the respective performance measures (Hamming accuracy, macro F1, and micro F1) across 10 different data splits. Therefore, some results are not statistically significant (i.e., ties, particularly among the baselines). However, we have double-checked using paired sample t-test with significance level of 0.05 and confirmed that (i) our three proposed methods strictly outperform all baselines, (ii) the LLM-based baselines (zero-shot and few-shot methods) strictly outperform other baselines.
>
> > W4 - Only binary class predictions
>
> We believe binary class prediction aligns with most applications in longitudinal human behavior modeling, where the goal is often to diagnose certain health/well-being outcomes such as depression, stress, and problematic use of various drugs. Therefore, we only considered binary class prediction for all tasks in this work. Conceptually, because our methods extend SummaryBoost (which considered single-label binary or multiclass classification) to MLC, extending them to multiclass MLC (where some or all of the Q labels are multiclass) is straightforward, as mentioned earlier in Footnote 6. We will consider extending our methodology and experiments in future work.
>
> ## Questions
>
> > 1. How do the authors combine data descriptions across topics?
>
> For clarifications, in the first step of our data conversion prompt, when the number of topics is reasonably small as illustrated in Figure S4b (newly added based on your review), we fit all topics within the data conversion prompt, hence there is only one data description for each record. When there are many topics, we split them into groups and respectively call the LLM to describe each group, resulting in multiple data descriptions. Then, in the second step, we append these data descriptions together (i.e., separated by newlines) to form a complete data description of the record.
>
> > 2. For the two psychology datasets (LifeSnaps and GLOBEM), why are some tasks (e.g., negative emotions or anxiety) defined differently across datasets?
>
> Following your review, we have clarified the factors deciding our label definitions for all considered datasets in Appendix C.1. For GLOBEM, we defined the label for PANAS task following the definition from (Englhardt et al. 2024), which is grounded on domain literature. For LifeSnaps, which studied participants in a different geographical region (Europe instead of U.S.), we simply defined the PANAS task for demonstration purposes only given the lack of conclusive findings regarding PANAS cutoffs in the area.
>
> > 3. When describing baselines in Section 4.2, what do problem transformation and algorithm adaptation mean?
>
> We would like to categorize the baselines given the two main approaches for MLC, which were first introduced in Section 2 (under MLC subsection).
>
> ### Minor comments / suggestions
>
> > Throughout I recommend using parenthetical citations (citep instead of cite)
> > Line 081: “have the two following”
> > Line 206: “require” instead of “requires”
>
> We have incorporated these helpful suggestions in our revised paper.
>
> > At multiple points, footnotes referenced in the main text were placed in the appendix. Could each footnote be placed on the page they are first referenced on?
>
> Due to the page limit, we resort to placing some footnotes in the appendix (only when they are mentioned in both the main text and the appendix).

---

> > ### Comment · Reviewer_qBMd · 2024-11-23
> > **Response to authors' reply**
> >
> > Thank you to the authors for responding to my concerns and questions. As stated in W1, my main critique was that the authors do not provide enough background clarification about the methods they build on (e.g. SummaryBoost). The authors' response was that they didn't include a detailed background on SummaryBoost because this method was built for a distinct application. While this may be true, as a reader who didn't have prior experience with the SummaryBoost method, having this background would make the methods description in the paper (section 3) much easier to parse. For this reason, I am choosing to keep my score as is (a weak accept).

---

> > > ### Author Response · Authors · 2024-11-24
> > >
> > > We appreciate Reviewer qBMd for your concern regarding the background clarification in our paper. We agree that this is necessary for readers who are unfamiliar with SummaryBoost, which is an important piece in our approach. Following your suggestion, we have included a subsection providing some background on SummaryBoost in Appendix D.7. We would like to keep the section labels intact to avoid confusion in this rebuttal, hence we will reorganize this background information (such that it appears first in the appendix) after the end of the current discussion period.
> > >
> > > Please let us know if the background we provided is sufficient. We are happy to include any further clarification.

---

> > > > ### Comment · Reviewer_qBMd · 2024-11-27
> > > > **thank you!**
> > > >
> > > > Thank you to the authors for providing an additional clarification paragraph on SummaryBoost. This was very helpful! Space permitting, I recommend the authors consider adding this paragraph (or parts of it) to the main text. I will raise my score.

---

> > > > > ### Author Response · Authors · 2024-11-27
> > > > >
> > > > > We sincerely thank Reviewer qBMd again for the time spent reviewing our paper! Your comments and suggestions have significantly improved the overall quality of our work. We will definitely consider adding the background information on SummaryBoost in the main text of our final version.

---

### Official Review · Reviewer_8yZu · 2024-10-27

**Soundness:** 3
**Presentation:** 2
**Contribution:** 2
**Rating:** 5
**Confidence:** 2

**Summary:**

This paper introduces MuHBoost, a multi-label boosting approach for predicting health and well-being outcomes from longitudinal human behavior data. MuHBoost is designed to address challenges associated with heterogeneous data, missing values, and computational demands in existing models. By incorporating LLM prompting with multi-label classification, MuHBoost aims to predict multiple health outcomes concurrently. Additionally, two variants of MuHBoost are proposed to manage potential hallucinations in LLMs, potentially enhancing model robustness. Evaluated on 13 prediction tasks using four health datasets, MuHBoost reportedly outperforms baseline models across multiple metrics, with resource efficiency highlighted as a key feature

**Strengths:**

* Demonstrates how LLMs can be used as multi-label boosting method for temporal tabular data and auxiliary tabular data.

**Weaknesses:**

* Iterative improvements on prior SummaryBoost work with limited novelty
	* The new data conversion uses a seemingly straightforward two-step approach based on LLM prompting in Section 3.1. The prompting itself does not seem to be very innovative, and while Table 2 does demonstrate an empirical performance increase, it is unclear how exactly each step of the two step approach helped.
	* MuHBoost modifies the original ClusterSampling algorithm by emphasizing rarer classes in the sampling procedure, then the follow-up methodologies MuHBoost[LP+] and MuHBoost[CC] seem to be straightforward adaptations of prior methods from Nam et al., 2017 and Adaboost.C2.
	* Prior work section does not seem to contextualize the importance and relevance of SummaryBoost, especially in context of the longitudinal ubiquitous computing domain.

**Questions:**

Comments:
* \cite is incorrectly used throughout the paper when \citep should be used

---

> ### Author Response · Authors · 2024-11-23
>
> We thank Reviewer 8yZu for taking the time to review our paper. We would like to respectfully present our rebuttal as follows.
>
> ## Weaknesses
>
> > Iterative improvements on prior SummaryBoost work with limited novelty
>
> While we indeed based our approach on SummaryBoost, it is not immediately clear how a tabular data classification method for small datasets can be applied to our problem of TSC given ubiquitous health data. Because this general form of time-series data may have high heterogeneity in practice (which has yet to be addressed by prior works as discussed in Appendix A), we leverage SummaryBoost to circumvent this challenge given its efficacy in heterogeneous tabular data classification and employment of LLMs (proven to be suitable for longitudinal human behavior modeling). Thus, we believe the novelty of our work comes mainly from the approach, rather than the individual techniques.
>
> > The new data conversion uses a seemingly straightforward two-step approach based on LLM prompting in Section 3.1. The prompting itself does not seem to be very innovative, and while Table 2 does demonstrate an empirical performance increase, it is unclear how exactly each step of the two step approach helped.
>
> Following up on our previous point, while the data conversion procedure is based on prompting techniques from existing works in various fields, it can effectively deal with general time series with high heterogeneity and dimensionality. Moreover, it can accommodate any auxiliary background data from the records, which can also be heterogeneous and high-dimensional (e.g., PWUD dataset). To our best knowledge, this flexibility is unseen in relevant works, which highlights the novel contribution of our data conversion approach.
>
> Regarding the individual performance of our two-step procedure, because the second step requires the data description(s) in complete sentences as input, we can only ablate either (i) the second step or (ii) both steps altogether. Based on your review, we have extended our ablation in Appendix C.3, under **Further Ablation**. As shown in Table S11, ablating (ii) results in a sharp decline in predictive performance compared to ablating (i), which implies more impact from the first step.
>
> > MuHBoost modifies the original ClusterSampling algorithm by emphasizing rarer classes in the sampling procedure, then the follow-up methodologies MuHBoost[LP+] and MuHBoost[CC] seem to be straightforward adaptations of prior methods from Nam et al., 2017 and Adaboost.C2.
>
> Although the two MuHBoost variants are based on existing works, we believe our work is the first to adapt their methodologies to LLMs. For (Nam et al., 2017), which reformulates MLTC problems and redesigns RNN architectures for generic text labeling, we adapted their idea by redesigning the inference directive in the inference prompt of LLMs, without the need for architectural design. Albeit straightforward, our approach effectively reduces the computational complexity of the problem and hence may alleviate hallucinations from LLMs as empirically shown in our newly added **MuHBoost vs. MuHBoost[LP+]** under Appendix C.3.
>
> For (Li et al., 2023), we apply the idea of combining AdaBoost with classifier chains (using traditional classifiers i.e., SVM for building weak learners) to enable LLM-based boosting classifier chains, which is novel to our best knowledge. Because this transforms MLC problems into multiple binary classification problems, our adaptation brings an alternative way of employing LLMs for MLC without the need for prompting multiple related questions simultaneously, which risks hallucinations especially for complex prediction tasks. Overall, we respectfully stand by our view that despite being straightforward and built upon prior works, the proposed methods are still novel in the sense of how they approach our (nontrivial and understudied) problem of interest in a simple-yet-effective way.
>
> > Prior work section does not seem to contextualize the importance and relevance of SummaryBoost, especially in context of the longitudinal ubiquitous computing domain.
>
> For clarification, SummaryBoost was proposed to specifically address tabular data classification with small sample sizes and high heterogeneity. Therefore, we do not include it in the related work as it cannot be applied to the longitudinal ubiquitous computing domain. We previously clarified its relevance to ubiquitous health data in the Introduction (in Contribution I). This unique approach to longitudinal human behavior modeling is hence the main novelty of our work.
>
> ## Questions
>
> > \cite is incorrectly used throughout the paper when \citep should be used
>
> We appreciate Reviewer 8yZu for noticing our inadvertent mistake. We have taken this into account in our revised paper.

---

> ### Author Response · Authors · 2024-11-25
>
> Dear Reviewer 8yZu,
>
> As the end of the discussion period approaches, we would like to check in with you on whether we have addressed your concerns of our paper. We are happy to discuss further if that is not the case.
>
> Sincerely,
>
> Authors of the current ICLR submission

---

> ### Author Response · Authors · 2024-11-27
>
> Dear Reviewer 8yZu,
>
> While we will not be able to upload our revised PDF after today (November 27 UTC), we can still address any of your further concerns. Please let us know if there is anything else we can provide to improve our paper.
>
> Sincerely,
>
> Authors of the current ICLR submission

---

### Official Review · Reviewer_aJj3 · 2024-10-31

**Soundness:** 3
**Presentation:** 3
**Contribution:** 2
**Rating:** 6
**Confidence:** 3

**Summary:**

This paper proposes a LLM-based method for multi-label classification, with focus on longitudinal human behavior. Built on SummaryBoost, this paper aims to address two limitations in the existing framework. First, the data could be heterogeneous in terms of data type (e.g., continuous measurements such as heart rate, and categorical responses, such as EMA data). Second, minimize the consumption of computing, time etc for inference. The proposed MuHBoost can handle the heterogeneous data type and enable efficient MLC. At the same time, two variants are proposed to mitigate hallucinations from LLMs. Extensive numerical experiments are conducted to evaluate the performance of the proposal, showing its potential advantages in longitudinal human behavior.

**Strengths:**

1. The paper identifies some significant issues of the existing LLM-based prediction methods, especially when dealing with multi-labels in longitudinal human behavior data.

2. The paper is well-written, and easy to follow, even if I do not have much research experience in this area.

3. The proposed enhancement for MuHBoost is helpful for small sample size problem, where N < 2^Q.

**Weaknesses:**

1. I don't think the proposed method can handle realistic high-dimensional time series, for example, movement, heart rate data collected from smartwatch. In the numerical experiments, the highest dimensionality of consideration is less than 30. Traditional machine learning, such as (Zhang et al., 2024) can handle very high frequency time series data (seconds, minutes), which can capture the human behavior in much finer level. Therefore, I think it is essential to discuss the limitations of the proposed method for high-frequency time series data, or justify the performance in numerical experiments, for example, authors can test the proposal on LifeSnaps dataset with much finer data. In addition, the authors may compare the proposal with the aforementioned method for prediction. If it is difficult to handle high-frequency time series data in the current proposal, authors may discuss how their method can be extended to those realistic settings.

Reference: Zhang, J., Xue, F., Xu, Q., Lee, J. and Qu, A., 2024. Individualized dynamic latent factor model for multi-resolutional data with application to mobile health. Biometrika, p.asae015.

2. I think the enhancement of MuHBoost in Section 3.3 lack novelties. Two approaches are largely motivated by two papers mentioned in the text. Specifically, LP+ is motivated by (Nam et al., 2017) while CC relies on the AdaBoost.C2 (Li et al., 2023). Please emphasize the novelties differentiated from these papers. Emphasize its benefits when adapting these approaches to the MuHBoost.

**Questions:**

1. In the original SummaryBoost paper, the numerical results show that SummaryBoost actually perform worse than traditional Xgboost when datasets have many continuous features. I think this is also the case for longitudinal human behavior data, such as heart rate. Did authors observe this phenomenon in the experiments?

2. Following the question 1, what are the input for MLkNN and MLTSVM? are they original measurements or embedding of text information or something else?

3. I think it worth comparing the proposed method with some traditional machine learning algorithms dealing with longitudinal data, such as RNN-type methods.

---

> ### Author Response · Authors · 2024-11-23
>
> We thank Reviewer aJj3 for your thorough review of our paper. We would like to respectfully present our rebuttal in the following.
>
> ## Weaknesses
>
> 1. For clarification, in our paper, we refer "high-dimensional time series" to those having a large number of features recorded over time and "long time series" to those consisting of many time points (i.e., either collected at high frequency or over a long period of time). While we indeed only conducted experiments on datasets with at most 30 time points, our methods can generally deal with long time series via the data conversion procedure as clarified in Appendix D.4 following your review. Unfortunately, we are not able to extend our experiments to longer time series due to the limitations of currently available ubiquitous health datasets, and hence we acknowledge this limitation from our work and have mentioned it in Appendix D.6.
>
> 2. Although the two MuHBoost variants are based on the two stated existing works, we believe our work is the first to adapt their methodologies to LLMs. For (Nam et al., 2017), which reformulates MLTC problems and redesigns RNN architectures for generic text labeling, we adapted their idea by redesigning the inference directive in the inference prompt of LLMs, without the need for architectural design. Albeit straightforward, our approach effectively reduces the computational complexity of the problem and hence may alleviate hallucinations from LLMs as empirically shown in our newly added **MuHBoost vs. MuHBoost[LP+]** under Appendix C.3. For (Li et al., 2023), we apply the idea of combining AdaBoost with classifier chains (using traditional classifiers i.e., SVM for building weak learners) to enable LLM-based boosting classifier chains, which is novel to our best knowledge. Because this transforms MLC problems into multiple binary classification problems, our adaptation brings an alternative way of employing LLMs for MLC without the need for prompting multiple related questions simultaneously, which risks hallucinations, especially for complex prediction tasks. Overall, the novelty of our methods comes mainly from the approach rather than the individual techniques.
>
> ## Questions
>
> 1. Recall that SummaryBoost was designed specifically for tabular data (Manikandan et al., 2023) and does not directly apply to longitudinal human behavior data. On the other hand, XGBoost can be applied to both forms of data. Therefore, it is impossible to compare SummaryBoost and XGBoost in our settings. However, following our results from Table 1, our methods still outperform XGBoost on LifeSnaps and GLOBEM, which contain mostly numerical features (in both time-series and auxiliary data). This can be attributed to the fact that while XGBoost works especially well for tabular data (Borisov et al., 2022), the same may not apply to longitudinal human behavior data, highlighting the contributions of our work.
>
> 2. As mentioned in Appendix C.2 when describing our baselines, for MLkNN and MLTSVM, we embedded the data descriptions via OpenAI’s embeddings API, which are then fed into these classifiers as input features. This is consistent with how the work of SummaryBoost (the foundation of our work) processed the data descriptions for their kNN baseline.
>
> 3. Following your suggestion, we have included another baseline based on RNN (Du et al., 2021), which was recommended by the authors of GLOBEM, and found that its performance is substantially worse than vanilla MuHBoost (and hence the two MuHBoost variants as well). Please refer to **Additional Benchmarking** in Appendix C.3 for further details.
>
> ## References
> (Borisov et al., 2022) Deep Neural Networks and Tabular Data: A Survey.
>
> (Du et al., 2021) AdaRNN: Adaptive Learning and Forecasting for Time Series.

---

> > ### Comment · Reviewer_aJj3 · 2024-11-24
> >
> > Thanks for clarifying all my questions and I really appreciate your efforts. I have the following comments:
> >
> > 1. given that there is no experiments for "real-world" long time series (very high-frequency time series like heart beat), I am not convinced that the proposed method can accommodate that very well. Or at least authors should state in the main text about this limitation, otherwise, it's misleading for readers that this paper can handle those real challenges.
> >
> > 2. My major concern about LLM for predictive problems (not limited to longitudinal modeling) is as follows: for continuous numerical data, you must do some conversion, like binning, quantiles, to input into LLM, which will loss some information compared with other machine learning method. For example, if we binarize multiple continuous variables, the correlation among those variables are totally messed up, which may lead to inaccurate downstream tasks. In the presented results, it seems this impact is not very significant, but I concern about its general application.
> >
> > Based on the above reasons, I will keep my score, and glad to hear more from authors.

---

> > > ### Author Response · Authors · 2024-11-25
> > >
> > > We thank Reviewer aJj3 for the follow-up comments and would like to address them in the following.
> > >
> > > 1. We agree with your concern about the proposed methods. First, we would like to point out our specific goal of longitudinal human behavior modeling in order to better position our contributions in the appropriate context. In this paper, we focus on modeling longitudinal human behaviors (e.g., substance use and mental health) that change in small periods (e.g., days) under heterogeneity (i.e., non-numerical with high rate of missing values) and potentially high dimensionality of the data. We believe this setting is realistic for applications involving early detection (i.e., binary classification) of certain health/well-being outcomes such as depression, stress, and problematic use of various drugs. On the other hand, works that specifically address long time series such as (Zhang et al., 2024) as you mentioned earlier focus on time-series interpolation, which is outside the scope of our paper. Regardless, we acknowledge the lack of supporting experiments for long time series in our work and will clearly state this limitation in the main text.
> > >
> > > While we do not focus on long time series, our methods can be adapted to said settings via data preprocessing (i.e., via our data conversion procedure). As mentioned in Section 3.1 and Appendix D.4 from our revised paper, this is achieved by segmenting the multivariate time series into multiple subperiods and then respectively prompting the LLM to describe each. We notice that datasets containing a large number of variables (i.e., high dimensionality as what we call in our paper), all or most of which are collected at high frequency, are extremely costly in real-world applications. Typically, only a small number of variables (such as heart rate) can be collected at fine scales. This applies to LifeSnaps, GLOBEM, and the real dataset employed in (Zhang et al., 2024) (which only includes five variables). In such cases, because the goal is to capture the overall pattern of each time-varying variable (e.g., heart rate follows specific rhythms), we can treat each finely recorded variable as a separate 'topic' within the data conversion procedure (assuming the number of those variables in the dataset is reasonably small). Given that LLMs have been empirically shown to be capable of zero-shot identifying patterns in general sequences (Gruver et al., 2023), we believe such accommodation leveraging the flexibility of our data conversion procedure can effectively deal with the long time series issue in practice.
> > >
> > > 2. We understand your concern regarding the employment of LLMs for real-world applications. As mentioned in both version of our paper (Line 1066 in the revised one), we do not discretize numerical data, i.e., by binning them into percentiles, as SummaryBoost originally did in the experiments. This is motivated by the fact that LLMs (e.g., GPT-3+) can already effectively zero-shot identify patterns in general sequences of raw numbers (Gruver et al., 2023), as well as the interpretability concerns for general health-related applications. Therefore, our data conversion procedure ensures minimal information loss for numerical time-series data, which is reflected in the good predictive performance of our methods on the LifeSnaps and GLOBEM datasets (consisting of mostly numerical features).
> > >
> > > ## References
> > > (Zhang et al., 2024) Individualized dynamic latent factor model for multi-resolutional data with application to mobile health.
> > >
> > > (Gruver et al., 2023) Large Language Models Are Zero-Shot Time Series Forecasters. NeurIPS-23.

---

> > > > ### Comment · Reviewer_aJj3 · 2024-11-25
> > > >
> > > > I greatly appreciate authors' timely responses, which address all my concern. I will raise the score accordingly.

---

> > > > > ### Author Response · Authors · 2024-11-25
> > > > >
> > > > > We sincerely thank Reviewer aJj3 again for the time spent reviewing our paper! Your helpful comments and suggestions have significantly improved the overall quality of our work.

---

### Official Review · Reviewer_wDHz · 2024-11-03

**Soundness:** 3
**Presentation:** 3
**Contribution:** 3
**Rating:** 8
**Confidence:** 4

**Summary:**

The paper proposes a novel approach named MuHBoost to model longitudinal human behavior. It is based on large language models (LLM) and aims to address the limitations of prior related works: inadequate consideration of realistic data type and the consumption of computing resources.

**Strengths:**

1. the proposal of MuHBoost as a resource-efficient multi-label classification method holds promise for solving the problem of multi-outcome health prediction.
2. The inclusion of both time series and ancillary data demonstrates the versatility of dealing with heterogeneous data formats common in health behavior modeling.
3. Comparative analyses show that MuHBoost generally outperforms benchmark methods on a variety of datasets, demonstrating its effectiveness.

**Weaknesses:**

1.The limited sample size of some datasets may affect the generalizability of the findings. This issue is only briefly acknowledged.
2. Scalability issues remain, especially in terms of the computational resources required when using MuHBoost for larger datasets or a wider range of applications.
3. The ideas presented in the article should be supported by more detailed derivation of the principle formulas.

**Questions:**

1.	In Section 1, the authors need more explicit narration of how their model tackles the three challenges presented. Adding a clearer distinction between MuHBoost and previous LLM-based approaches would help readers better understand the unique contributions of the model.

2.	In Section 2, most of the Multi-Label Classification related works are a bit out-of-date, more recent works (2023/2024) should be included.

3.	In Section 3, “the increased computational burden due to longer and more complex prompts may lead to hallucinations from LLMs”needs more solid proof. The hallucination of LLM is a widely and deeply studied topic, the authors haven’t provided clearly (experimentally/theoretically) how their proposed method addresses this issue.  For section 3.1: The data transformation process could be graphically illustrated to make it clearer to readers who are not familiar with data aggregation techniques.

4.	In Section 4, some of the compared baselines are too old (e.g., Random Forest and XGBoost). The state-of-the-art results are not convincing enough.

5.	In Section 4 and Appendix C.3, the resource consumption problem is not fully elaborated. The time complexity analysis needs further proof. In addition, more experimental indicators (e.g., FLOPS) should be included to prove the efficiency of the proposed method.

---

> ### Author Response · Authors · 2024-11-23
>
> We thank Reviewer wDHz for taking the time to review our paper. We would like to first comment on the weaknesses and then address your questions.
>
> ## Weaknesses
>
> 1. Recall that our main goal is to develop methodologies that mainly consider the three stated challenges of ubiquitous health data (and two shortcomings of state-of-the-art i.e., LLM-based approaches). These challenges include small sample size due to the typically high data acquisition cost. You are correct that this brings a limitation to our work. Based on your suggestions, we have included a Limitation section in Appendix D.6. More specifically, we do not expect our proposed methods to scale to larger datasets (i.e., with thousands of samples) in terms of resource efficiency. When the opportunity to work with large ubiquitous health data arises, approaches that require ample sample size such as LLM finetuning become more suitable than ours i.e., boosting LLM-generated weak learners, which is also the case for SummaryBoost.
>
> 2. Yes, you are correct. Following up on our previous point, our approach focuses on ubiquitous health datasets that typically come with small sample sizes. Therefore, we do not claim the scalability of our developed methods for large datasets. Nevertheless, we believe the proposed idea of employing LLMs for MLC in real-world health prediction tasks is novel (as discussed in related work from Appendix B) and could be applied to significantly reduce resource consumption for data-extensive approaches such as finetuning (which is prohibitively expensive when the number of health/well-being outcomes to predict, Q, is large).
>
> 3. We apologize for any confusion in our presentation. During writing, we tried to follow the outlines and technical details provided in the SummaryBoost paper from NeurIPS-23, which forms the foundation of our methodology. In the paper, the authors presented their work with only algorithms and illustrations, and hence we adopted this convention. We are willing to elaborate on our derivations if Reviewer wDHz could provide more details on which "principle formulas" are in question.
>
> ## Questions
>
> 1. Based on your suggestions, we have further clarified how our proposed methods address the three challenges of ubiquitous health data compared to existing LLM-based approaches in Appendix D.4.
>
> 2. Our current literature review has already provided a thorough coverage on MLC, as we are not aware (as of writing this rebuttal) of any innovative work from established venues in recent years. For MLTC, we have included our updated literature review. Please refer to the last paragraph in Appendix B for a discussion on more recent works.
>
> 3. Based on your suggestion, we have provided more solid proof. In particular, in Table S10 (newly added from your review), we show that MuHBoost underperforms MuHBoost[LP+], particularly for PWUD dataset where the number of labels Q is larger. For further details, please refer to **MuHBoost vs. MuHBoost[LP+]** in Appendix C.3.
>
> > The hallucination of LLM is a widely and deeply studied topic, the authors haven’t provided clearly...
>
> Our approach actively alleviates LLM hallucinations during (1) the data conversion procedure and (2) the inference process (via the two MuHBoost variants). For (1), we demonstrated its positive effect through our ablation study Impact of Refining Data Description in Section 4.3. To further support our claim based on your review, we have conducted another ablation study where we omit the entire data conversion procedure (Table S11). Please refer to **Further Ablation** in Appendix C.3. For (2), please refer to **MuHBoost vs. MuHBoost[LP+]** in the same appendix for how our two MuHBoost variants can mitigate hallucinations from LLMs.
>
> > For section 3.1: The data transformation process could be graphically illustrated...
>
> Based on your suggestion, we have included an illustrative example in Figure S4 of our revised paper.
>
> 4. In existing longitudinal human behavior modeling studies, all state-of-the-art approaches were compared against these baselines. Therefore, we naturally considered them as baselines following the convention. We also compared our methods to those state-of-the-art approaches. Furthermore, based on your review, we have included another baseline (Du et al., 2021), which was recommended by the authors of GLOBEM as a potential solution to longitudinal human behavior modeling. Our results in Table S12 show that MuHBoost still outperforms this baseline. Please refer to **Additional Benchmarking** in Appendix C.3 for further details.
>
> 5. Based on your suggestions, we have included an analysis of time complexity in Appendix D.5. Because it may take arbitrary time to call LLMs (e.g., intermittent delays), we do not report empirical runtime in our experimental results (see also Footnote 16). We have also included a paragraph on FLOPs estimation in Appendix D.5.
>
> ## Reference
> (Du et al., 2021) AdaRNN: Adaptive Learning and Forecasting for Time Series.

---

> ### Author Response · Authors · 2024-11-25
>
> Dear Reviewer wDHz,
>
> As the end of the discussion period approaches, we would like to check in with you on whether we have addressed your concerns of our paper. We are happy to discuss further if that is not the case.
>
> Sincerely,
>
> Authors of the current ICLR submission

---

> > ### Comment · Reviewer_wDHz · 2024-11-25
> >
> > Dear Authors,
> >
> > I have carefully reviewed all the reviewers' comments and your responses. Since these comments and responses have resolved my concerns, I will not repeat any questions.
> >
> > Have a nice day :)

---

> ### Author Response · Authors · 2024-11-27
>
> Dear Reviewer wDHz,
>
> We are glad to hear that we have addressed your concerns. Again, we greatly appreciate your helpful review of our paper! Your comments and suggestions have significantly improved the overall quality of our work.
>
> Sincerely,
>
> Authors of the current ICLR submission

---

### Author Response · Authors · 2024-11-23
**Revised Paper Uploaded**

Dear Reviewers,

We greatly appreciate your thorough reviews of our paper, which significantly improve the overall quality of the proposed work. In response to your helpful comments and suggestions, we have uploaded our revised paper to this portal. Please note that when we say "please refer to Appendix XYZ" in our rebuttal, we are referring to the revised version.

Sincerely,

Authors of the current ICLR submission

---

### Meta-Review · Area_Chair_qMf6 · 2024-12-22

**Metareview:**

This paper proposed an efficient LLM-based multi-label classification method over longitudinal and spare data. The method was evaluated  on 13 tasks from 4 datasets, mainly on longitudinal human behaviors (eg substance use, mental health).
One of the focus of the downstream tasks is to classify changes with minimal seen examples and sparse longitudinal data.

The method is built on SummaryBoost, in order to address two limitations. First, the data could be heterogeneous in terms of data type (e.g., continuous measurements such as heart rate, and categorical responses, such as EMA data).
Second, the data doesn't have to be large scale, in which LLM fine-tuning can be more useful. In this case, the authors adapted an existing method - SummaryBoost, and specifically Cluster Sampling in their framework.
Thus, the method is quite versatile as it is able to handle data heterogeneity.

The discussion between authors and reviewers was robust. Almost all the reviewers increased their scores during the rebuttal.
The biggest weakness of this paper is the adaptation of existing methods -- and the seemingly simple prompting mechanisms. This major concern was raised by one of the reviewers and seems that the authors have not fully addressed their concerns.

in my view is this paper is quite well executed, with a very comprehensive set of experiments and benchmarks. The idea may seemingly be 'too simplistic' - however, perhaps that is all that is needed for the tasks at hand.

**Additional Comments On Reviewer Discussion:**

One reviewer championed the paper for acceptance, whereas another reviewer was quite adamant that the work has limited novelty given that it's only adapted existing method -- eg Cluster Sampling, SummaryBoost, into an LLM-based method, specifically with just prompting.

This paper is just above the borderline range, however, the major concern requires SAC to weigh in their views, as the reviewers are not unanimous in their recommendation.
We need SAC to weigh in - whether the contributions are deemed enough for ICLR.

---

### Decision · Program_Chairs · 2025-01-22

Accept (Poster)